# Detection of isoforms and genomic alterations by high-throughput full-length single-cell RNA sequencing in ovarian cancer

Arthur Dondi[1,2,40], Ulrike Lischetti [1,39,40] ✉, Francis Jacob [3], Franziska Singer [2,4], Nico Borgsmüller[1,2], Ricardo Coelho[3], Tumor Profiler Consortium*, Viola Heinzelmann-Schwarz[3,23], Christian Beisel [1] ✉ & Niko Beerenwinkel [1,2] ✉

Understanding the complex background of cancer requires genotype-phenotype information in single-cell resolution. Here, we perform long-read single-cell RNA sequencing (scRNA-seq) on clinical samples from three ovarian cancer patients presenting with omental metastasis and increase the PacBio sequencing depth to 12,000 reads per cell. Our approach captures 152,000 isoforms, of which over 52,000 were not previously reported. Isoform-level analysis accounting for non-coding isoforms reveals 20% overestimation of protein-coding gene expression on average. We also detect cell type-specific isoform and poly-adenylation site usage in tumor and mesothelial cells, and find that mesothelial cells transition into cancer-associated fibroblasts in the metastasis, partly through the TGF-β/miR-29/Collagen axis. Furthermore, we identify gene fusions, including an experimentally validated *IGF2BP2::TESPA1* fusion, which is misclassified as high *TESPA1* expression in matched short-read data, and call mutations confirmed by targeted NGS cancer gene panel results. With these findings, we envision long-read scRNA-seq to become increasingly relevant in oncology and personalized medicine.

Cancer is a complex disease characterized by genomic and transcriptomic alterations[1] that drive multiple tumor-promoting capabilities or hallmarks[2]. Among others, these alterations include point mutations, insertions and deletions (indels), and gene fusions on the genomic level, and splice isoforms on the transcriptomic level. Their detection offers great potential for personalized oncology as they can serve as direct therapeutic targets[3,4] or potential neoantigens informing on the immunogenicity of the tumor[5]. Gene fusions arising from large-scale genomic rearrangements, for example, play an oncogenic role in a variety of tumor types[6], and are successfully used as

therapeutic targets[7,8]. Like mutations[9] and copy number variations[10], fusion rates can vary widely across cancer types, and gene fusions are thought to be drivers in 16.5% of cancer cases, and even the only driver in more than 1%[11]. Furthermore, out-of-frame gene fusions are more immunogenic than mutations and indels, making them an ideal target for immunotherapies and cancer vaccines[12,13]. On the transcriptomic level, alternative splicing is a major mechanism for the diversification of a cell's transcriptome and proteome[14] and can impact all hallmarks of tumorigenesis. It also presents a non-genomic source of potential neoantigens[15]. In breast and ovarian cancer, 68% of samples had at least

[1]ETH Zurich, Department of Biosystems Science and Engineering, Mattenstrasse 26, 4058 Basel, Switzerland. [2]SIB Swiss Institute of Bioinformatics, Mattenstrasse 26, 4058 Basel, Switzerland. [3]University Hospital Basel and University of Basel, Ovarian Cancer Research, Department of Biomedicine, Hebelstrasse 20, 4031 Basel, Switzerland. [4]ETH Zurich, NEXUS Personalized Health Technologies, Wagistrasse 18, 8952 Schlieren, Switzerland. [39]Present address: University Hospital Basel and University of Basel, Ovarian Cancer Research, Department of Biomedicine, Hebelstrasse 20, 4031 Basel, Switzerland. [40]These authors contributed equally: Arthur Dondi, Ulrike Lischetti. *A list of authors and their affiliations appears at the end of the paper. ✉e-mail: ulrike.lischetti@unibas.ch; christian.beisel@bsse.ethz.ch; niko.beerenwinkel@bsse.ethz.ch

one isoform detected in proteomic data with an exon-exon junction that was not previously reported in the literature (neojunction)[16].

The complexity of cancer further extends to intra-tumor heterogeneity[17] and its intricate interplay with the tumor microenvironment (TME)[18]. Ultimately, to fully decipher functional tumor heterogeneity and its effect on the TME, single-cell resolution providing both phenotype and genotype information is required. Single-cell RNA sequencing (scRNA-seq) is now widely used for the phenotypic dissection of heterogeneous tissues. It can be divided into short-read, high-throughput technologies allowing for gene expression quantification and long-read, low-throughput technologies that cover full-length isoforms[19]. Up to now, short- and long-read methods had to be used in parallel to combine the advantages of each technology. The long-read sequencing field is rapidly expanding[20], with scRNA-seq methods being constantly developed and improved on Nanopore[21,22] and PacBio[23-27] platforms. So far, long-read RNA-seq has however only been applied on the bulk level in the field of oncology[25,28,29]. High-quality, high-throughput, long-read scRNA-seq has the potential to provide isoform-level cell-type-specific readouts and capture tumor-specific genomic alterations. With near ubiquitous p53 mutations and defective DNA repair pathways causing frequent non-recurrent gene fusions, high-grade serous ovarian cancer (HGSOC) is an ideal candidate to investigate these alterations[10,30,31].

Here, we used high-quality, high-throughput long-read scRNA-seq to capture cell-type-specific genomic and transcriptomic alterations in clinical cancer patients. We applied both short-read and long-read scRNA-seq to five samples from three HGSOC patients, comprising 2571 cells, and generated the PacBio scRNA-seq dataset with the deepest coverage to date. We were able to identify over 150,000 isoforms, of which a third were not previously reported, as well as cell-type-specific isoforms. Isoform-level analysis revealed that, on average, 20% of the protein-coding gene expression was noncoding, leading to an overestimation of the protein expression. By combining differential isoform and polyadenylation site usage analysis between cells from the metastatic TME and distal omental biopsies, we found evidence that in omental metastases, mesothelial cells transition into CAFs, partly through the TGF-β/miR-29/Collagen axis. Additionally, we discovered dysregulations in the insulin-like growth factor (IGF) network in tumor cells on the genomic and transcriptomic levels. Thereby, we demonstrated that scRNA-seq can capture genomic alterations accurately, including cancer- and patient-specific germline and somatic mutations in genes such as *TP53*, as well as gene fusions, including an *IGF2BP2::TESPA1* fusion.

## Results

### Long-read scRNA-seq creates a catalog of isoforms in ovarian cancer patient-derived tissue samples

We generated short-read and long-read scRNA-seq data from five omentum biopsy samples (Supplementary Tables 1 and 2) from three HGSOC patients. Three samples were derived from HGSOC omental metastases and two from matching distal tumor-free omental tissues (Fig. 1a). To generate long reads, we opted for the PacBio platform for its generation of high-fidelity (HiFi) reads through circular consensus sequencing (CCS). To overcome its limitations in sequencing output and optimize for longer library length, we (1) removed template-switch oligo artifacts that can account for up to 50% of reads through biotin enrichment, (2) concatenated multiple cDNA molecules per CCS read, and (3) sequenced on the PacBio Sequel II platform (2–4 SMRT 8M cells per sample, "Methods"). This allowed the generation of a total of 212 Mio HiFi reads in 2571 cells, which, after demultiplexing, deduplication, and artifact removal, resulted in 30.7 Mio unique molecular identifiers (UMIs), for an average of 12k UMIs per cell (Supplementary Table 1). There was a mean of four cDNA molecules concatenated per sequencing read overall, and cDNA length was similar across samples (Supplementary Fig. 1a, b). Artifact removal filtered 51% of the

reads, and it included the removal of intrapriming (63%), noncanonical isoforms (36%), and reverse-transcriptase switching (1%)[32] (Supplementary Fig. 1c). It must be emphasized that those artifact reads emerge from the single-cell library preparation and are also present in short-read data, where they cannot be filtered and are hence accounted for as valid reads.

The long-read dataset revealed 152,546 isoforms, each associated with at least three UMIs. We classified the isoforms according to the SQANTI classification[32] and calculated their proportions ("Methods" and Fig. 1b, c): full splice match (FSM)−isoforms already in the GENCODE database (32.8%), incomplete splice matches (ISM)−isoforms corresponding to shorter versions of the FSM (35.1%), novel in catalog (NIC)−isoforms presenting combinations of known splice donors and acceptors (15.9%), and novel not in catalog (NNC)−isoforms harboring at least one unknown splice site, or neojunction (14.4%). Novel isoforms (classes NIC and NNC) accounted for 30% of the isoforms, and 11% of the total reads in all samples, while FSM accounted for 33% of the isoforms and 80% of the reads (Fig. 1c, d), indicating that high coverage is required for the reliable detection of new, low-abundant, transcripts.

To evaluate the structural integrity of all isoforms, we compared their 5′ end to the FANTOM5 CAGE database[33] and their 3′ end to the PolyASite database[34] (Fig. 1e). More than 82% of the NIC and 74% of NNC isoforms could be validated on 3′ and 5′ ends, similarly to FSM. As expected, fewer ISM isoforms were found to be complete (42%): they are either incompletely sequenced isoforms missing their 5′ end (30%) or the result of early 3′ termination (55%). FSM, NIC, and NNC had overall better 3′ and 5′ validation than the full-length tagged isoforms in the GENCODE database (Fig. 1e). Only the 'Matched Annotation from NCBI and EMBL-EBI' (MANE[35]) containing curated representative transcripts cross-validated between the GENCODE and RefSeq database had a better 3′ and 5′ validation of 95%. A total of 52,884 novel isoforms were complete (NIC+NNC), of which 40,046 were confirmed by GENCODE as valid isoforms not previously reported (corresponding to 17% of the current GENCODE v36 database), and 3695 were extended versions of existing isoforms. Isoforms that were not confirmed were mainly either "partially redundant with existing transcripts", or "overlapping with multiple loci". Finally, we assessed the presumed functional categorization (biotypes) of novel isoforms. We found that 42% are protein-coding, more than the 36% of protein-coding isoforms found in the GENCODE database (230k entries) (Fig. 1f, g). This demonstrates the ability of concatenated long-read sequencing to generate high-yield, high-quality data and characterize isoforms that were not previously reported.

### Long-read sequencing allows for short-read-independent cell-type identification

Next, through comparison to short-read data, we assessed the ability of long-read sequencing to cluster cells and to identify cell types. We generated short- and long-read gene count matrices and removed non-protein-coding, ribosomal, and mitochondrial genes. After filtering, we obtained 16.5 Mio unique long reads associated with 12,757 genes, and 26.3 Mio unique short reads associated with 13,122 genes (Supplementary Table 2). The short- and long-read datasets were of similar sequencing depth with a median of 4930 and 2750 UMIs per cell, respectively (average 10,235 and 6413 UMIs, Supplementary Fig. 2a). Long-read data contained slightly fewer detected genes, and genes detected in both datasets overlapped by 86.4% (Supplementary Fig. 2b, c). Paradoxically, the genes detected were overall shorter in long reads than short reads, likely due to the concatenation step (Supplementary Fig. 2d).

We first identified cell types independently per cell, using cell-type marker gene lists ("Methods" and Supplementary Fig. 3a). We compared short- and long-read data and found that both data types identified cell types with very similar percentages, namely HGSOC (13.4% in short-read vs 13.6% in long-read data), mesothelial cells (20.2 vs 20.5%),

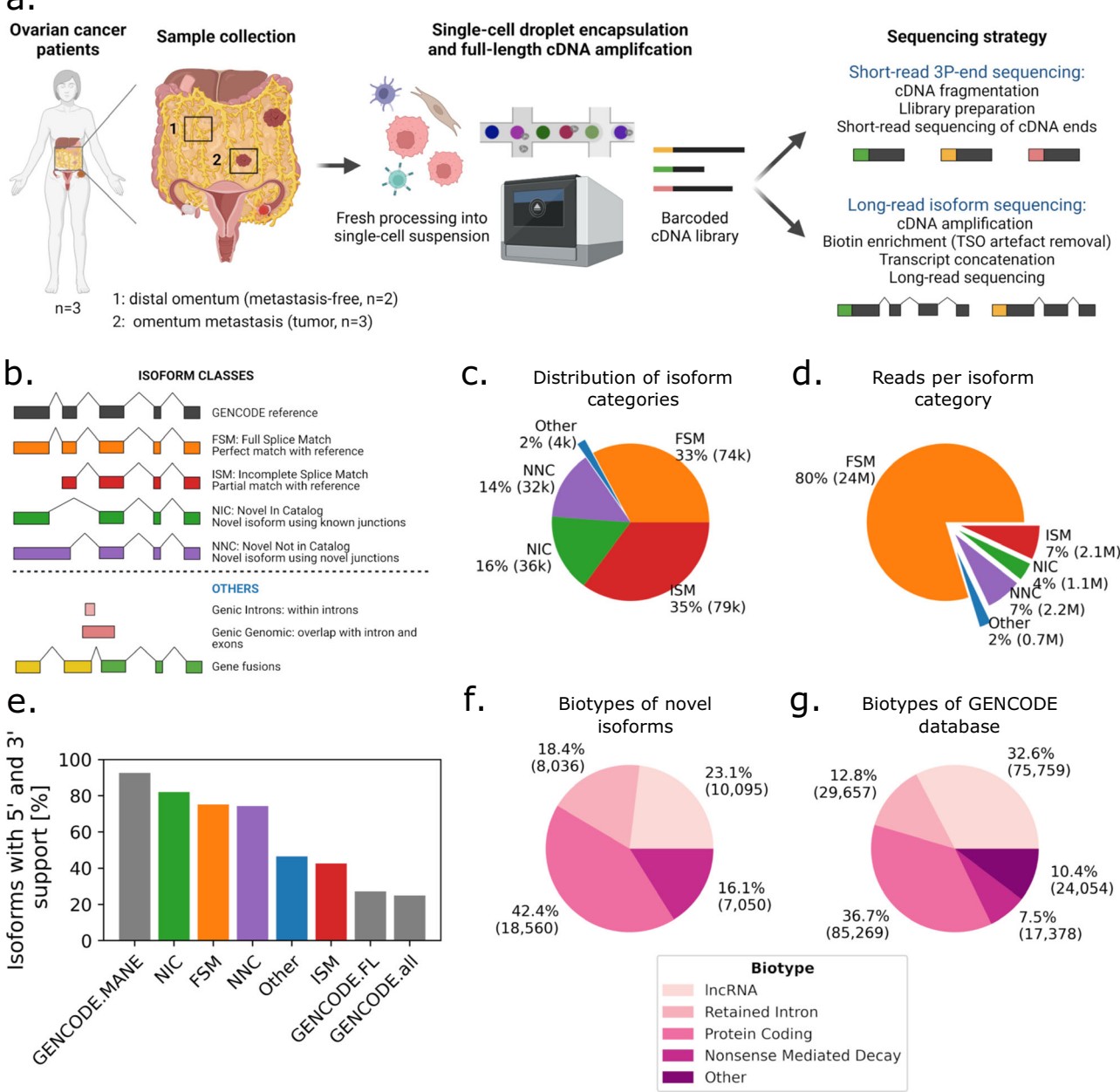

**Fig. 1 | Study design and long-read data overview. a** Schematic of freshly processed HGSOC omentum metastases and patient-matched tumor-free distal omentum tissue biopsies, scRNA-seq. **b** Definition of SQANTI-defined isoform structural categories. **c** Proportions of isoform structural categories detected in merged metastasis and distal omentum samples. Percentage and total number of isoforms per category are indicated. **d** Proportions of unique reads attributed to isoforms detected in (**c**). Percentage and total number of UMIs per category are indicated. **e** Percentage of isoforms for which transcription start site is supported by CAGE (FANTOM5) data and transcription termination site is supported by polyadenylation (PolyASite) data, per isoform structural categories. "GENCODE.all" indicates all protein-coding isoforms in the GENCODE database, "GENCODE.FL" is a subset of 'GENCODE.all' containing only isoforms tagged as full-length, and "GENCODE.MANE" is a hand-curated subset of canonical transcripts, one per human protein-coding locus. **f** GENCODE-defined biotype composition of novel isoforms. **g** Biotype composition of the GENCODE database.

fibroblasts (10.7 vs 10.7%), T cells (38.7 vs 38.5%), myeloid cells (14.6 vs 14.9%), B cells (1.4 vs 1.1%), and endothelial cells (1.1 vs 1.4%) (Supplementary Fig. 3b). We then projected short-read gene, long-read gene, and long-read isoform expression onto 2-dimensional embeddings using UMAP[36] (Fig. 2a). We manually clustered cell types based on the embeddings and calculated the Jaccard distance between clusters. Cell clusters based on short- and long reads were very similar, with a Jaccard distance >94% for all cell types except B cells, where the Jaccard distance was >75% (Fig. 2b). Furthermore, Jaccard similarity analysis between cell-type clusters and attributed cell-type labels were analogous between short- and long-read data, with a better prediction of B

cells and endothelial cells for long reads (Supplementary Fig. 3c). Except for tumor cells, cell types did not show patient-specific clustering (Supplementary Fig. 3d).

**Long-read sequencing captures germline and somatic mutations and identifies increased neojunctions in tumor cells**
Next, we assessed the potential of long-read data for mutation detection, and used somatic mutations to further validate the cell-type annotation. Germline mutations are expected in all cell types, whereas somatic mutations should be present only in tumor cells. As reference, we used mutations called from a panel covering 324 genes on

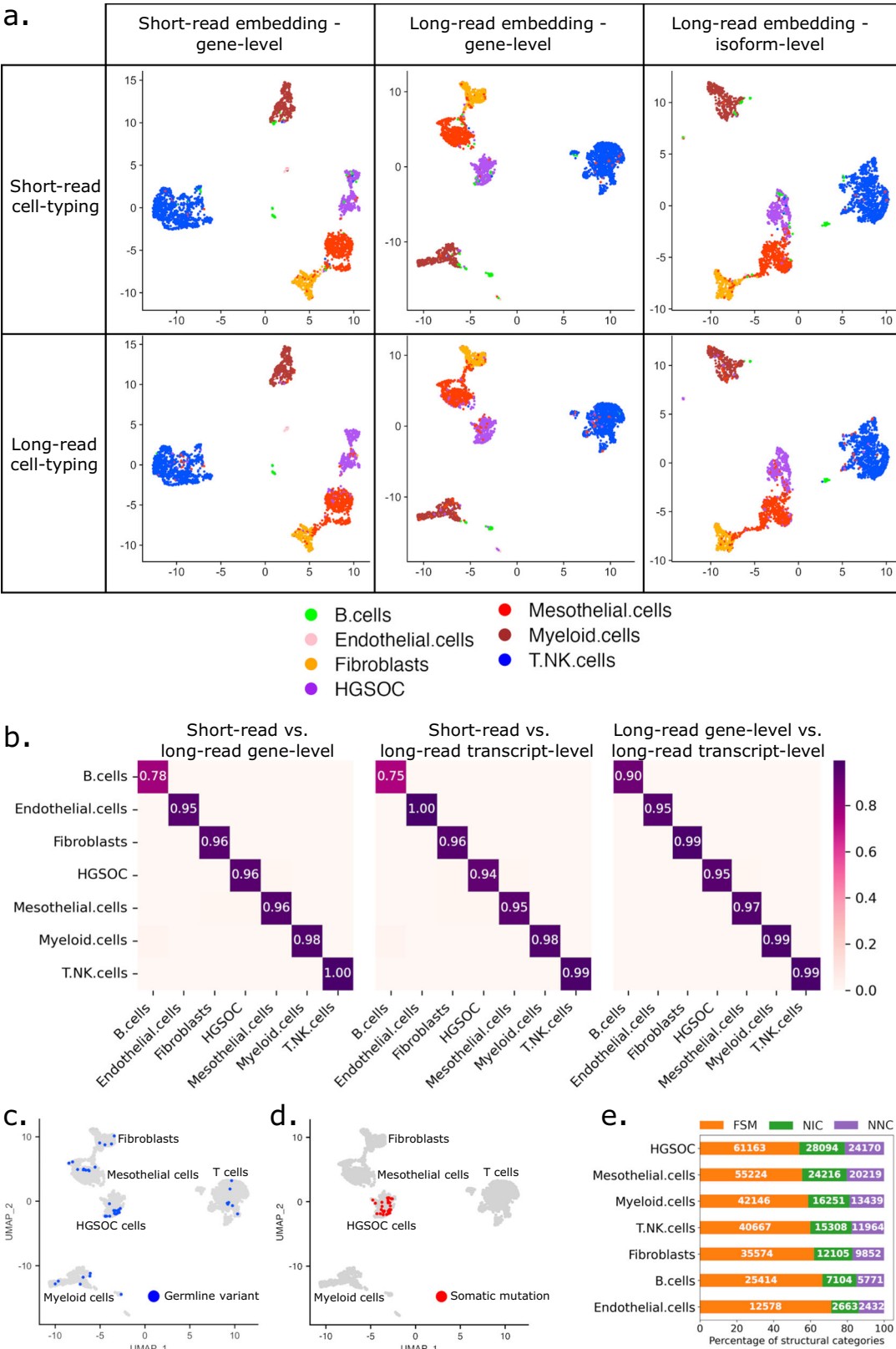

**Fig. 2 | Clustering and cell-type-specific isoform distribution. a** Cohort UMAP embeddings by data types and automatic cell-type annotation. Top and bottom rows: cell-type labels based on short- and long-read data, respectively. Left column: embedding on short-read data−gene level, middle column: embedding on long-read data−gene level, right column: embedding on long-read data−isoform level. **b** Jaccard distance of cell populations in different UMAP embeddings: short reads−gene level versus long reads− gene level (left), short reads−gene level versus long reads−isoform level (middle), long reads−gene level versus long reads−isoform level (right). Long reads−gene-level UMAP cohort visualizations of cells with at least one germline (**c**) or somatic (**d**) mutation also found in targeted NGS panel data of matched patient samples. Germline variants are variants detected in healthy omentum distal samples. **e** SQANTI-defined structural category normalized distribution of isoforms detected per cell type (number of isoforms displayed in white).

patient-matched bulk DNA samples ("Methods"). We identified germline variants in 48 cells belonging to all cell types from the distal omentum and tumor sites (Fig. 2c and Supplementary Data 1). Somatic mutations were called in 34 cells, all in the cell cluster annotated as tumor cells (Fig. 2d). In 20 of those cells, *TP53* was found mutated (Supplementary Data 1). Thus, high-fidelity, long-read data can be leveraged for both germline and somatic mutation calling.

Cell-type-specific isoform expression analysis (overall and per gene) revealed increased isoform and transcript expression in HGSOC cells compared to other cell types (Supplementary Fig. 4a–c). This difference does however not translate into mean UMIs per isoform, as isoforms expressed in cancer cells harbor fewer UMIs than in mesothelial cells (Supplementary Fig. 4d). This means that cancer cells express more low-abundant isoforms suggesting wider isoform diversity and broader cellular functions, and controls. Isoform class distribution between cell types revealed a higher fraction of novel isoforms and neojunctions (NNC) in tumor cells (Fig. 2e). Looking into unique isoform expression in the different cell types, we found that cancer cells contained more than 8% (9476) of cell-type-specific isoforms, and 35% of those contained neojunctions (NNC) ("Methods", Supplementary Fig. 4e). This was an increase of 2.3–10.6 times compared to the other cell types. At the cellular level, 0.5% of the cancer-specific isoforms were also unique to a single cell, which is between three and six times the percentage of unique isoforms in other cell types, and 50% contained neojunctions (Supplementary Fig. 4f). Those rare isoforms were difficult to detect with previous methods,. Taken together, cancer cells expressed at least twice as many unique isoforms than other cell types, indicating an increased transcriptomic diversification and supporting previous findings of cancer-specific neojunction expression in bulk data[16].

## Tumor microenvironment shows epithelial-to-mesenchymal transition through TGFβ-driven miR-29 downregulation

In the subsequent analysis, we compared stromal fibroblasts and mesothelial cells derived from metastasis (TME) and matched tumor-free (distal) omentum. In distal samples, fibroblasts and mesothelial cells formed distinct clusters in both short- and long-read data (Fig. 3a, left and Supplementary Fig. 5a). In metastatic samples, however, TME fibroblasts and mesothelial cells formed a bridge in the UMAP embedding, suggesting that these cells might undergo a cell state transition. To test this hypothesis, we conducted gene set enrichment analysis and found that the epithelial-to-mesenchymal transition (EMT) pathway was enriched in TME compared to distal mesothelial and fibroblast cells (Fig. 3a, right). Similarly, collagen fibril organization and extracellular matrix (ECM) pathways were enriched in TME cells, indicating a reprogramming of the TME cells during metastasis formation (Supplementary Fig. 5b). In addition, we compared the alternative polyadenylation (APA) ("Methods"). Among the 2876 genes tested for APA, the isoforms of 26 genes in TME mesothelial cells exhibited significant 3'UTR lengthening compared to distal mesothelial cells, while 13 genes showed shortened 3'UTRs (Fig. 3b). Collagen-encoding genes *COL1A2*, *COL3A1*, *COL5A2*, and *COL6A1* were similarly lengthened and upregulated in TME cells (Fig. 3c), with *COL1A2* having the highest effect size ($P_{corr} = 3.42 \times 10^{-67}$, 53% change).

As 3'UTR lengthening is usually associated with a decrease in expression due to microRNA (miRNA) silencing[37], the increased usage of lengthened 3'UTR in TME mesothelial cells suggests that distal cells may express a distinct set of miRNAs not present in TME cells. Collagen-encoding genes are known to be regulated by the miR-29 family in fibroblasts[38]. Thus, we used miRDB[39] to predict gene targets for silencing by miR-29a/b/c (Supplementary Data 2). Among the 26 isoforms with lengthened 3'UTR, 9 were predicted targets of miR-29a, almost all described as EMT actors (collagens[40–43], *KDM6B*[44], *TNFAIP3*[45], *FKBP5*[46], and *RND3*[47]). In contrast, none of the 13 shortened 3'UTR isoforms were predicted to be miR-29 targets (Fig. 3b). Furthermore,

we compared gene expression of TME and distal mesothelial cells (Supplementary Data 3), and genes with lengthened 3'UTR isoforms predicted to be silenced by miR-29 were significantly overexpressed ($P = 1.35 \times 10^{-3}$) in TME mesothelial cells compared to the ones not predicted to be silenced by miR-29a (Fig. 3d). Mesothelial TME cells also differentially expressed miR-29 targets which are major ECM genes, such as collagen gene *COL1A1* (fold change=9.0, $P_{corr}=1.72 \times 10^{-124}$), *MMP2*[48] (fold change=4.1, $P_{corr}=1.51 \times 10^{-30}$), and *LOX*[49] (fold change=10.6, $P_{corr}=6.33 \times 10^{-51}$), which is also lengthened ($P_{corr}=7.07 \times 10^{-2}$, 77% change). Overall, ECM-related genes known to be targeted by miR-29 were upregulated in TME cells compared to distal cells (Fig. 3e), supporting the hypothesis that the observed EMT was potentially linked with the miR-29 downregulation. When comparing differentially expressed isoforms in TME mesothelial cells to distal cells, *COL1A1* was also the gene with the highest change in relative isoform abundance amongst all its isoforms ($P_{corr} = 6.34 \times 10^{-49}$, 86% usage change, "Methods") (Fig. 3f). In TME mesothelial cells, the *COL1A1* canonical 3' polyadenylation site was used, whereas distal cells used a premature polyadenylation site, leading to the formation of truncated isoforms (Fig. 3f). When incorporating only protein-coding isoforms and removing the truncated isoforms from the analysis, the gene expression fold change increased from 9 to 62-folds ($P_{corr}=3.15 \times 10^{-183}$). This overexpression of canonical *COL1A1* in the TME can be explained by the absence of miR-29 silencing, as previously described[50].

The miR-29 family is known to be an EMT inhibitor[50]. Its silencing through the TGFβ pathway correlates with the upregulation of ECM-encoding genes, including multiple collagens, as reported in the present study. The main TGFβ gene *TGFB1* was found to be enriched in TME mesothelial cells (fold change=1.4, $P_{corr} = 2.32 \times 10^{-2}$). Furthermore, in distal mesothelial cells, 38% of *TGFB1* isoforms comprise an alternative 3' exon, leading to aberrant protein expression (Supplementary Fig. 5c), while the canonical protein-coding *TGFB1* isoform *ENST00000221930.6* is overexpressed in TME cells (fold change=2.3, $P_{corr} = 6.59 \times 10^{-9}$). miR-29 is also regulated through the expression of noncoding RNAs that act as molecular sponges, directly binding to miR-29 and, therefore, leading to the overexpression of their targets. The TGFβ-regulated long noncoding RNA H19, which enhances carboplatin resistance in HGSOC, has been reported to promote EMT through the *H19*/miR-29b/*COL1A1* axis[51–53] and was found to be overexpressed in the TME mesothelial cells (fold change=4.6, $P_{corr} = 3.46 \times 10^{-34}$). Circular RNAs have also been described as miR-29 sponges, notably circMYLK, and circKRT7 in HGSOC. circMYLK and circKRT7 originate from *MYLK* and *KRT7*, respectively, which are both significantly overexpressed in TME mesothelial cells (fold change=4.1, $P_{corr}=6.97 \times 10^{-96}$ and fold change=4.6, $P_{corr}= 2.18 \times 10^{-13}$)[54,55]. Similarly, TME mesothelial cells expressed the endogenous isoform of *GSN* (cGSN), while distal cells only expressed the secreted isoform (pGSN)[56] (Supplementary Fig. 5d). *cGSN* has been shown to be under TGF-β control in breast cancer and to increase EMT marker expression[57]. In conclusion, our findings strongly support that, in omental metastases, the mesothelial cells transition into cancer-associated fibroblasts (CAFs), partly through the TGF-β/miR-29/Collagen axis.

## Differential isoform usage in cancer reveals changes in biotypes

After comparing cells from the TME with distal cells, we investigated which isoforms, biotypes, and polyadenylation sites were differentially used between cancer and all distal cells. HGSOC cells expressed isoforms differentially with a change in relative isoform abundance of more than 20% in 960 genes (15.1%), compared to all distal cells (6353 genes tested in total, Fig. 4a, Supplementary Data 4 and "Methods"). In 36% of those 960 genes, the highest expressed isoform biotype changed between conditions (Fig. 4b). In 32% of instances, there was a transition from a protein-coding to a non-protein-coding isoform, and in 17% of cases, cancer cells expressed a protein-coding isoform while

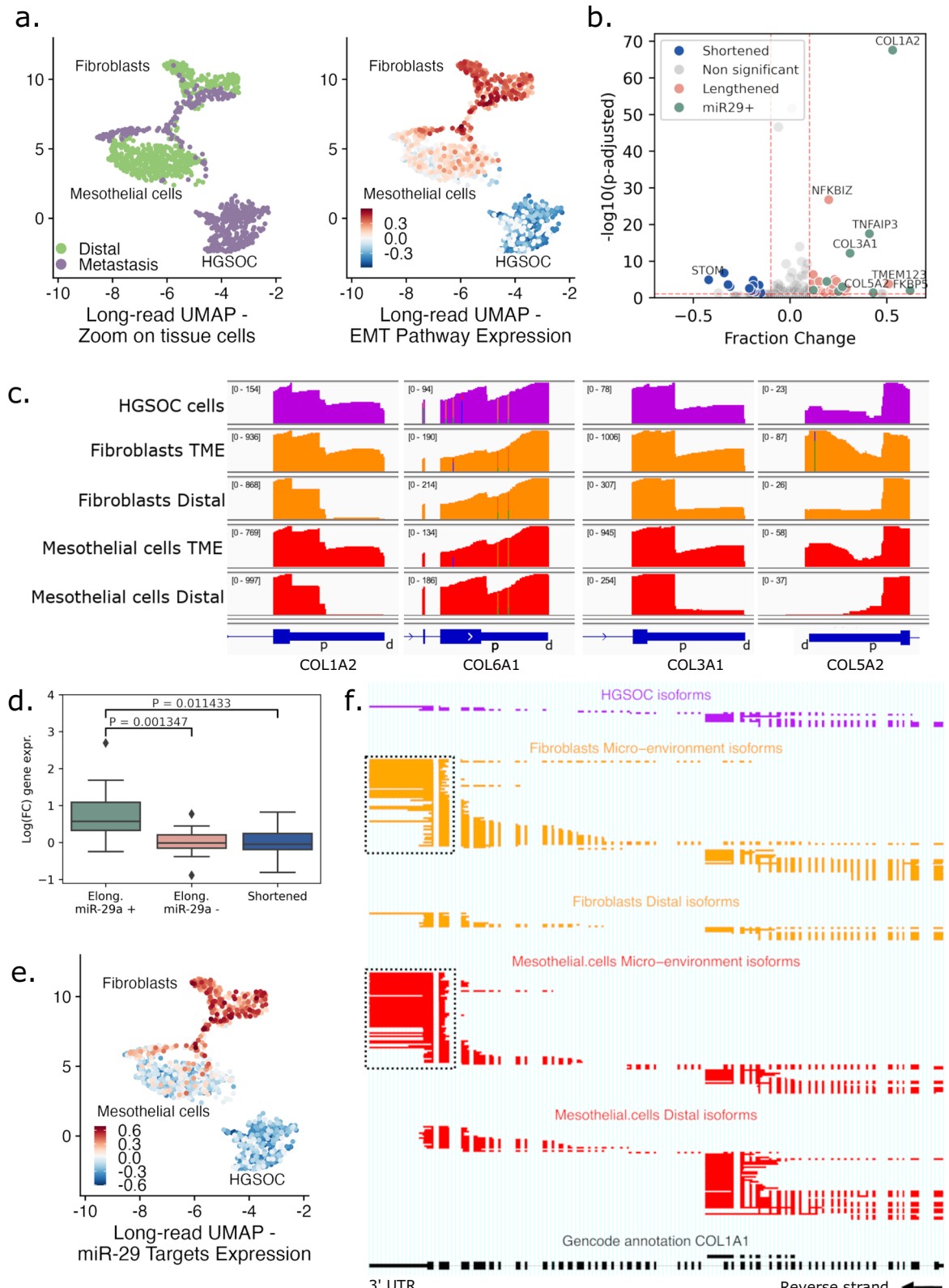

distal cells expressed a non-protein-coding isoform. Only 39 genes (0.6%) had an isoform switch with a change in relative isoform abundance of more than 50%. 59% of these switched isoforms demonstrated a biotype transition (49% of protein-coding to non-protein-coding transition, Fig. 4c), and in 33% of cases, cancer cells expressed a protein-coding isoform while distal cells expressed a non-protein-coding isoform. In addition, in cancer, distal, and TME cells, on average

20–21% of the expression in protein-coding genes was noncoding (Supplementary Fig. 6a), and 13–14% of protein-coding genes had more than 50% of noncoding expression. This means that, on average, using only gene-level information to estimate protein expression (as done in short-read data) will lead to an overestimation of 20%.

The ten genes with the statistically most significant switches were *IGF1, TPM2, NCALD, VAMP8-VAMP5, EXOSC7, ICAM3, CERK, OBSL1, GSN,*

**Fig. 3 | Epithelial-to-mesenchymal transition in the tumor microenvironment.**
**a** Zoom of UMAP embeddings of the cohorts' long-read–gene-level data (Fig. 2a, middle column) highlighting tumor and stromal (mesothelial and fibroblast) cells, colored by biopsy tissue type (left) and EMT gene set signal (right). **b** Volcano plot of genes with APA in mesothelial cells. Genes have either a lengthened (red) or shortened 3'UTR in TME compared to distal mesothelial cells. Differentially lengthened or shortened genes targeted by miR-29 are colored in green. Genes with -log10(p-adjusted) >10 and |Fraction Change| >0.4 are annotated. APA statistical test is described in "Methods" (**c**) IGV view of 3'UTR raw coverage of *COL1A2*, *COL6A1*, *COL3A1*, and *COL5A2* in tissue cell types. On the top left between brackets, the coverage range is displayed throughout each condition. In blue, Ensembl canonical 3'UTR, and for each gene, distal (d) and proximal (p) APA sites are

annotated. **d** Log fold-change expression between TME and distal mesothelial cells of lengthened genes targeted (+, green, n = 9) or not targeted (−, red, n = 12) by miR-29, and shortened genes (blue, n = 19). Boxes display the first to third quartile with median as horizontal line, whiskers encompass 1.5 times the interquartile range, and data beyond that threshold is indicated as outliers. *P* values were calculated using a two-sided Student's *t*-test between the fold-change means. **e** Cohort UMAP embedding long-read data–gene level, colored by gene set signal of ECM-related genes targeted by miR-29. **f** ScisorWiz representation of *COL1A1* isoforms. Colored areas are exons, whitespace areas are intronic space, not drawn to scale, and each horizontal line represents a single read colored according to cell types. Dashed boxes highlight the use of the canonical 3' UTR in TME fibroblasts and mesothelial cells, while distal mesothelial cells use an earlier 3' exon termination.

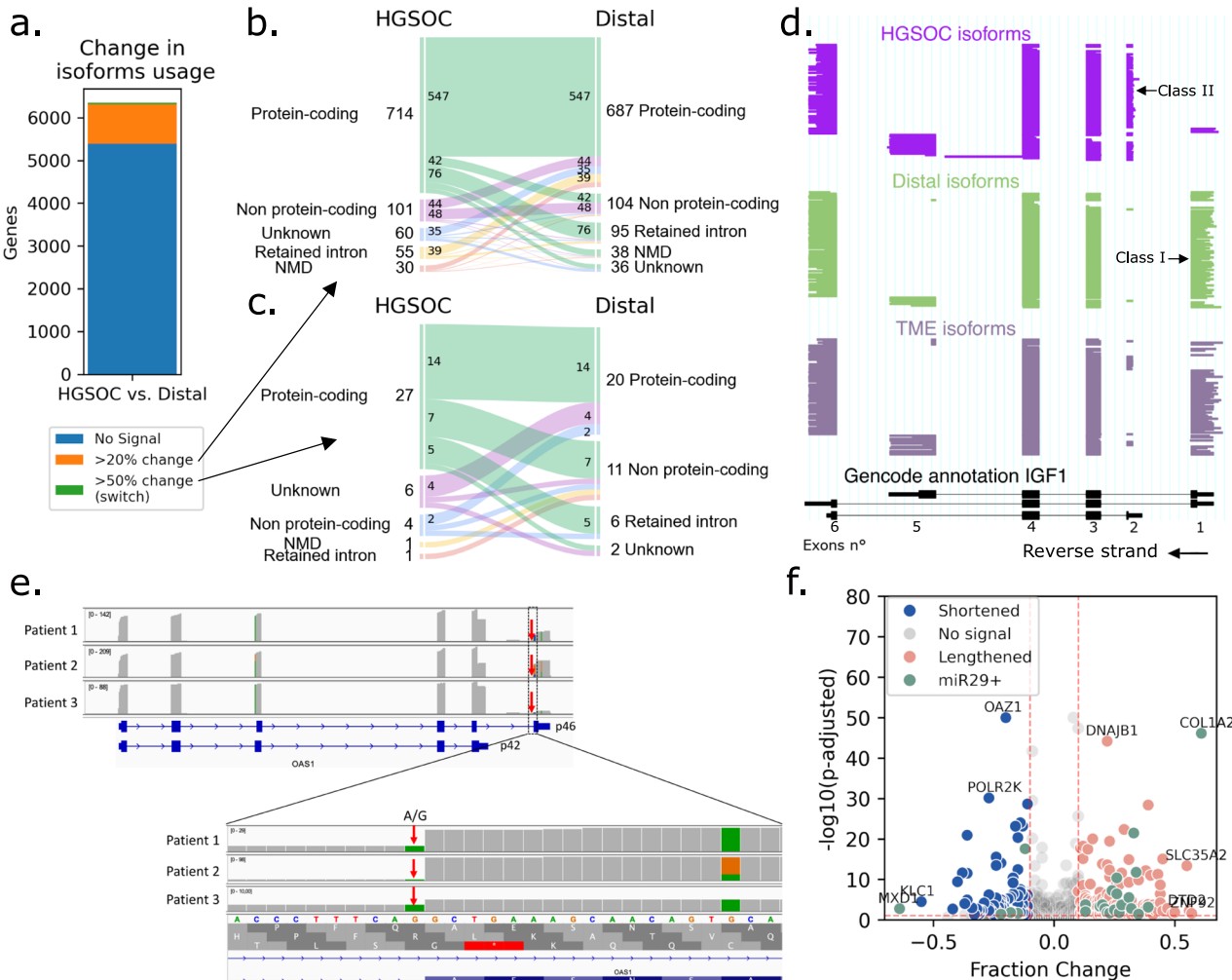

**Fig. 4 | Differential isoforms and 3'UTR lengths in cancer. a** Number of genes with change in isoform usage between HGSOC and all distal cells. In orange, genes with differentially expressed isoforms and a change in relative isoform abundance >20% (>50% in green). In blue, genes with no differentially expressed isoforms or change in relative isoform abundance <20%. **b** Alluvial plot of biotypes of most expressed isoforms in HGSOC and distal cells in genes containing an isoform change >20% (n = 960). Each vein represents the conversion of one biotype to another. For example, in seven genes, the most expressed isoform in HGSOC cells is protein-coding, while distal cells' one is non-protein-coding. **c** Alluvial plot of biotypes of most expressed isoforms in HGSOC and distal cells in genes containing an isoform switch (>50% change, n = 39). **d** ScisorWiz representation of isoforms in

*IGF1*, each horizontal line represents a single isoform colored according to cell types. Exons are numbered according to the Gencode reference, *Class I* and *II* isoforms are isoforms with starting exons 1 and 2, respectively. **e** Top: IGV view of *OAS1* expression in patients. Patient 3 has low p46 expression compared to others. Bottom: zoom on the last exon of isoform p46, where all patients have at least one mutated A allele in the splice acceptor site. **f** Volcano plot of genes with APA in cancer versus distal cells. Genes have either a lengthened (red) or shortened (blue) 3'UTR in cancer cells compared to all distal cells. Differentially lengthened or shortened genes targeted by miR-29 are colored in green. Genes with -log10(*P*-adjusted) >10 and |fraction change| >0.5 are annotated. APA statistical test is described in "Methods".

and *RTN1* (Supplementary Data 4 and "Methods"). In *IGF1*, cancer cells across all patients predominantly used the second exon of the gene as their transcription start site (secreted isoform, *Class II*), whereas non-cancerous cells primarily used the first exon (endogenous isoform, *Class I*)[58] (Fig. 4d). On the contrary, similarly to TME mesothelial cells, cancer cells expressed the endogenous isoform of *GSN* (*cGSN*), while distal cells only expressed the secreted isoform (*pGSN*)[56] (Supplementary Fig. 5d). In *RTN1*, distal cells expressed the isoform *RTN1-A* and *RTN1-B*, while Patient 3's cancer cells expressed *RTN1-C*, an isoform known to bind to the anti-apoptotic protein Bcl-xL and reduce its activity[59,60] (Supplementary Fig. 6b). In the tropomyosin gene *TPM2*, which is involved in TGF-β-induced EMT, cancer cells differentially expressed exon 6b (isoform *TPM2.3*, expressed in epithelial cells[61]) and the alternative 3'UTR exon 9a (Supplementary Fig. 6c). In *VAMP5*, the overexpressed isoform in HGSOC cells was a *VAMP8-VAMP5* transcriptional read-through, i.e., RNA formed of two neighboring genes on the same chromosome, previously described in human prostate adenocarcinoma[62] (Supplementary Fig. 6d). HGSOC cells expressed almost no wild-type (wt) *VAMP5* but had a significantly higher *VAMP8* expression than distal cells ($P_{corr}=1.0 \times 10^{-15}$), indicating that this read-through gene was under transcriptional control of *VAMP8*. With a short-read 3' capture method, this *VAMP8-VAMP5* expression cannot be distinguished from the wt *VAMP5* expression. For *NCALD* and *OBSL1*, only cancer cells expressed canonical protein-coding isoforms, while other cells expressed short noncoding isoforms (Supplementary Fig. 6e, f). For *CERK*, by contrast, Patient 2's HGSOC cells strongly expressed an isoform not previously reported, leading to a shortened protein (Supplementary Fig. 6g). Finally, in *ICAM3*, cancer cells mainly expressed a short protein-coding isoform, while distal cells (mainly T cells) expressed the canonical isoform. More characterization of those isoforms will be necessary in the future to explore the biological implications linked to their expression (Supplementary Fig. 6h).

Although isoforms differentially expressed in cancer cells were similar among patients, there was one significant case of patient-specific expression. For *OAS1*, Patient 3 predominantly expressed isoform *p42*, while Patients 1 and 2 exhibited a balanced distribution of isoforms *p42* and *p46*. The *p42* and *p46* expressions are known to be allele-specific, caused by the rs10774671 SNP, a splice acceptor A/G variation. However, this cannot explain the different expression levels as all patients have both the A and G alleles (Fig. 4e). Given that isoform *p42* is more susceptible to Nonsense-Mediated mRNA Decay (NMD) and possesses diminished enzymatic activity[63], the observed differences could potentially indicate a diminished OAS1 activity in Patient 3. Whether *OAS1* has an impact on ovarian cancer is still to be elucidated.

When testing for differential APA between HGSOC and distal cells, we found shortened 3'UTR in 85 and lengthened 3'UTR in 203 genes ($n = 4758$) (Fig. 4f). There was a notable trend toward lengthening of the 3'UTR in cancer cells ($P = 5.59 \times 10^{-20}$), with *COL1A2* emerging once more as the most prominent finding ($P_{corr}=7.48 \times 10^{-47}$, 61% change) (Figs. 3b, c and 4f). Expression levels remained consistent between genes featuring either shortened or lengthened 3'UTRs. Furthermore, neither miRNA profiles nor canonical pathways exhibited an overlap exceeding 20% with either the lengthened or shortened gene sets ("Methods").

**Long-read sequencing captures gene fusions and identifies an *IGF2BP2::TESPA1* fusion that was misidentified in short-read data**
To detect fusion transcripts, we aligned long reads to the reference genome and filtered for reads split-aligned across multiple genes. We then ranked fusion transcripts with counts across all cells of more than 10 UMIs (Supplementary Data 5). Out of the 34 detected fusion entries, 21 were genes fused with mitochondrial ribosomal RNA (*mt-rRNA1-2*) and ubiquitous among all cell types, 11 isoforms were *IGF2BP2::TESPA1* fusions specific to Patient 2, one was a cancer cell-specific *CBLC* (chr8:43.064.215) fusion to a long noncoding RNA (lncRNA) expressed

in Patient 3, and one was a cancer cell-specific fusion of *FNTA* with a lncRNA expressed in Patient 1. The ubiquitous *mt-rRNA* fusions were likely template-switching artifacts from the library preparation, as *rRNA* makes up to 80% of RNA in cells[64]. *IGF2BP2::TESPA1* was a highly expressed fusion event in Patient 2: 2174 long reads mapped to both *IGF2BP2* (Chr3) and *TESPA1* (Chr12). The gene fusion consisted of 5' located exons 1–4 of *IGF2BP2*, corresponding to 112 amino acids (aa) and including the RNA-recognition motif 1 (RRM1) and half of the RRM2 domain, linked to the terminal *TESPA1* 3' untranslated region (UTR) exon, encoding 69 aa as in-frame fusion and including no known domains (Fig. 5a). In total, the gene fusion encoded 181 aa, compared to 599 aa of wt *IGF2BP2* and 521 aa of wt *TESPA1* (Fig. 5b). 98.9% of fusion reads were found in HGSOC cells and the fusion was detected in 86.8% of Patient 2's cancer cells, making it a highly cancer cell- and patient-specific fusion event (Fig. 5c). Cancer cells lacking the gene fusion had lower overall UMI counts, suggesting low coverage as a possible reason for the absence of the gene fusion (Fig. 5d).

We next investigated the footprint of the gene fusion in the short-read data. The *TESPA1* gene was expressed in T cells, as well as in HGSOC cells, where its expression values were elevated. High expression was exclusive to Patient 2 HGSOC cells and colocalized with *IGF2BP2* expression (Fig. 5e, f). *TESPA1* was the highest differentially expressed gene in cancer cells compared to non-cancer cells in Patient 2 ($P_{corr} < 1.17 \times 10^{-14}$). Next, we re-aligned Patient 2's short reads to a custom reference including the *IGF2BP2::TESPA1* transcriptomic breakpoint as well as wt *TESPA1* and wt *IGF2BP2* junctions (Supplementary Fig. 7a and "Methods"). Out of the 994 reads mapping to the custom reference, 93% preferentially aligned to *IGF2BP2::TESPA1* (99.8% of those were from HGSOC cells). This means that, when given the option, reads previously aligning to *IGF2BP2* or *TESPA1* are preferentially mapping to the fusion reference, and the reported overexpression of *TESPA1* in short reads is likely an *IGF2BP2::TESPA1* expression. Furthermore, reads covering the *TESPA1* 3' UTR region harbored three heterozygous single nucleotide polymorphisms (hSNPs): chr12:54.950.144 A > T (rs1047039), chr12:54.950.240 G > A (rs1801876), and chr12:54.950.349 C > G (rs2171497). In long reads, wt *TESPA1* was either triple-mutated or not mutated at all, indicating two different alleles. All fusion long reads, however, were triple-mutated, indicating a genomic origin and monoallelic expression of the fusion (Fig. 5g). In short reads, the three loci were mutated in nearly all reads, supporting the hypothesis that the observed *TESPA1* expression represents almost completely *IGF2BP2::TESPA1* expression and that it has a genomic origin.

**Genomic breakpoint validation of the *IGF2BP2::TESPA1* fusion**
To validate that the *IGF2BP2::TESPA1* gene fusion is the result of genomic rearrangements, both bulk and single-cell DNA sequencing (scDNA-seq) data from matched omental metastasis was used to query the genomic breakpoint. A putative genomic breakpoint was first found in the RNA data. Two long-read fusions were mapped to intronic regions of *IGF2BP2* and *TESPA1* (Supplementary Fig. 7b), pinpointing the location of the breakpoint at position chr3:185,694,020–chr12:54,960,603. Subsequent genotyping PCR on genomic DNA extracted from patient-matched tissue samples using *IGF2BP2::TESPA1*, wt *IGF2BP2*, and wt *TESPA1* primer pairs flanking the genomic breakpoint confirmed the presence of *IGF2BP2::TESPA1* in Patient 2 in 3 out of 4 tested samples (Fig. 6a and "Methods"). In contrast and as expected, the fusion was not found in Patient 1.

To assess whether the fusion was exclusive to cancer cells, we further investigated scDNA-seq data from Patient 2. For the identification of cancer cells, we inferred the scDNA-seq copy number profiles of all cells. We successfully identified two distinct clones within the pool of 162 cells. These clones encompassed a cancer clone designated as "Subclone 0" and a presumably non-cancer clone without copy number alterations labeled as "Subclone 1" (Fig. 6b). We next aligned

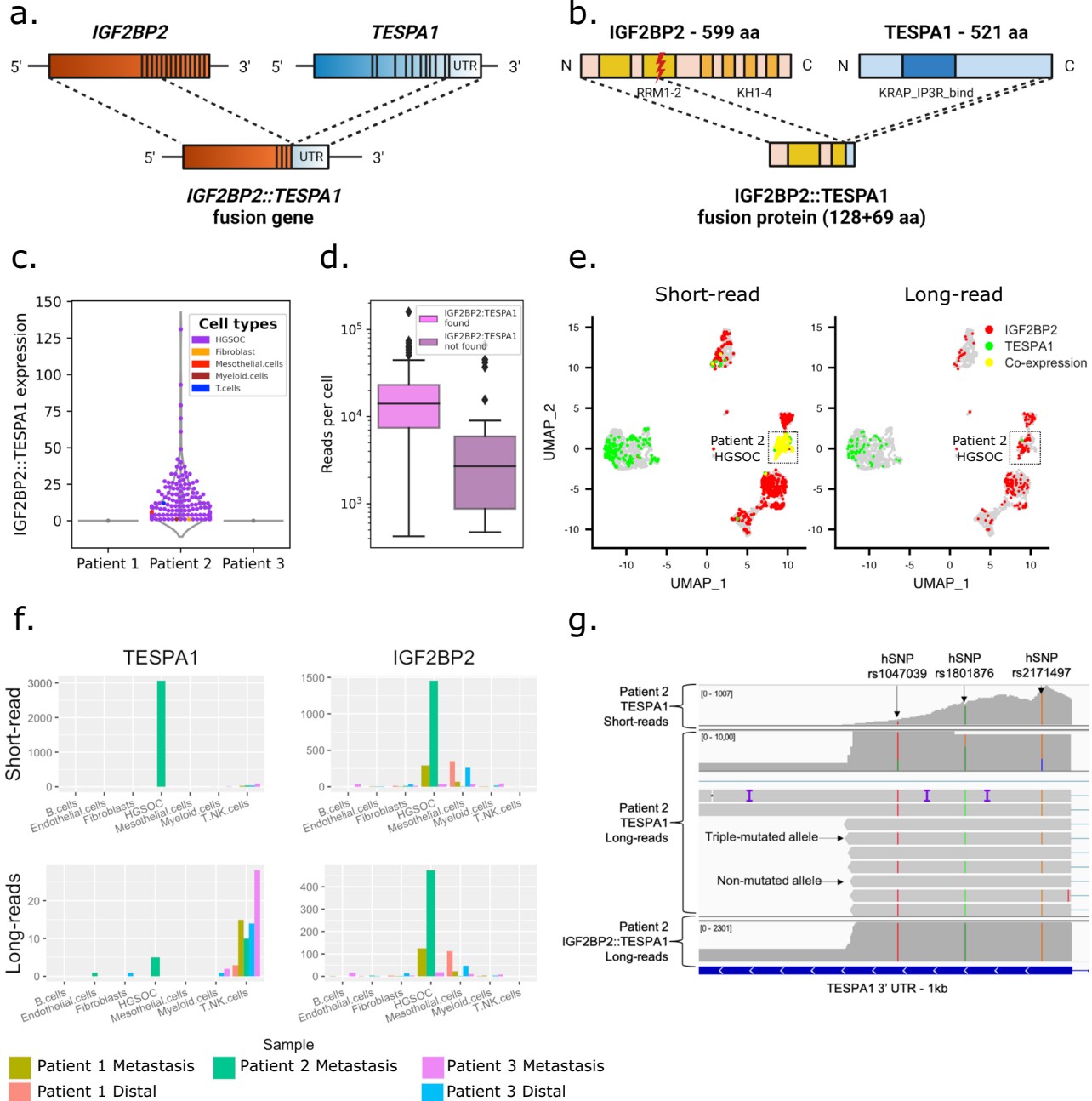

**Fig. 5 | Tumor and patient-specific detection of a *IGF2BP2::TESPA1* gene fusion.**
**a** Overview of wt *IGF2BP2*, wt *TESPA1*, and *IGF2BP2::TESPA1* gene fusion with exon structure. **b** Overview of wt IGF2BP2, wt TESPA1, and fusion protein with protein domains. RRM RNA-recognition motif, KH heterogeneous nuclear ribonucleoprotein K-homology domain, KRAP_IP3R_bind Ki-ras-induced actin-interacting protein-IP3R- interacting domain. **c** Violin plot showing patient- and tumor-specific *IGF2BP2::TESPA1* fusion transcript detection in Patient 2. **d** UMI count in fusion-containing (*n* = 173) versus -lacking (*n* = 32) Patient 2 tumor cells. Boxes display the first to third quartile with median as horizontal line, whiskers encompass 1.5 times the interquartile range, and data beyond that threshold is indicated as outliers. **e** UMAP embeddings of the cohorts' short-read data. Cells are colored if they express *IGF2BP2* (red), *TESPA1* (green), or both (yellow) in short- (left panel) or long reads (right panel). **f** Raw expression of *TESPA1* (left) and *IGF2BP2* (right) in short- (top) or long reads (bottom), by sample and cell type. **g** IGV view of short reads (top), non-fusion long reads (middle), and fusion long reads (bottom) mapping to the 3′UTR of *TESPA1*. Non-fusion reads are either triple hSNP-mutated or non-mutated, while fusion and short reads are only triple hSNP-mutated.

the scDNA-seq data to a custom reference covering the breakpoint ("Methods") and only found cancer reads mapping to the fusion breakpoint (P=0.032, two-sided Fisher's exact test), while a mixture of reads from cancer and non-cancer cells mapped to wt *IGF2BP2* and wt *TESPA1* (*P* = 0.78 and *P* = 1.00, respectively, two-sided Fisher's exact test) (Fig. 6c). Thus, genotyping PCR of bulk extracted DNA and scDNA-seq data confirmed the genomic fusion breakpoint in the intronic region detected in long-read scRNA-seq data. scDNA-seq also

confirmed that the *IGF2BP2::TESPA1* fusion was cancer cell-specific, as suggested by long-read scRNA-seq data.

*IGF2BP2* was also overexpressed in cancer cells from Patient 2 compared to other patients on both RNA and protein levels (Supplementary Fig. 8a–c). In Patient 2, there was an elevated copy number observed within the genomic region encompassing *IGF2BP2* (Fig. 6b). Therefore, the presence of a fusion allele on one allele does not seem to hinder the transcription of the wt *IGF2BP2* allele.

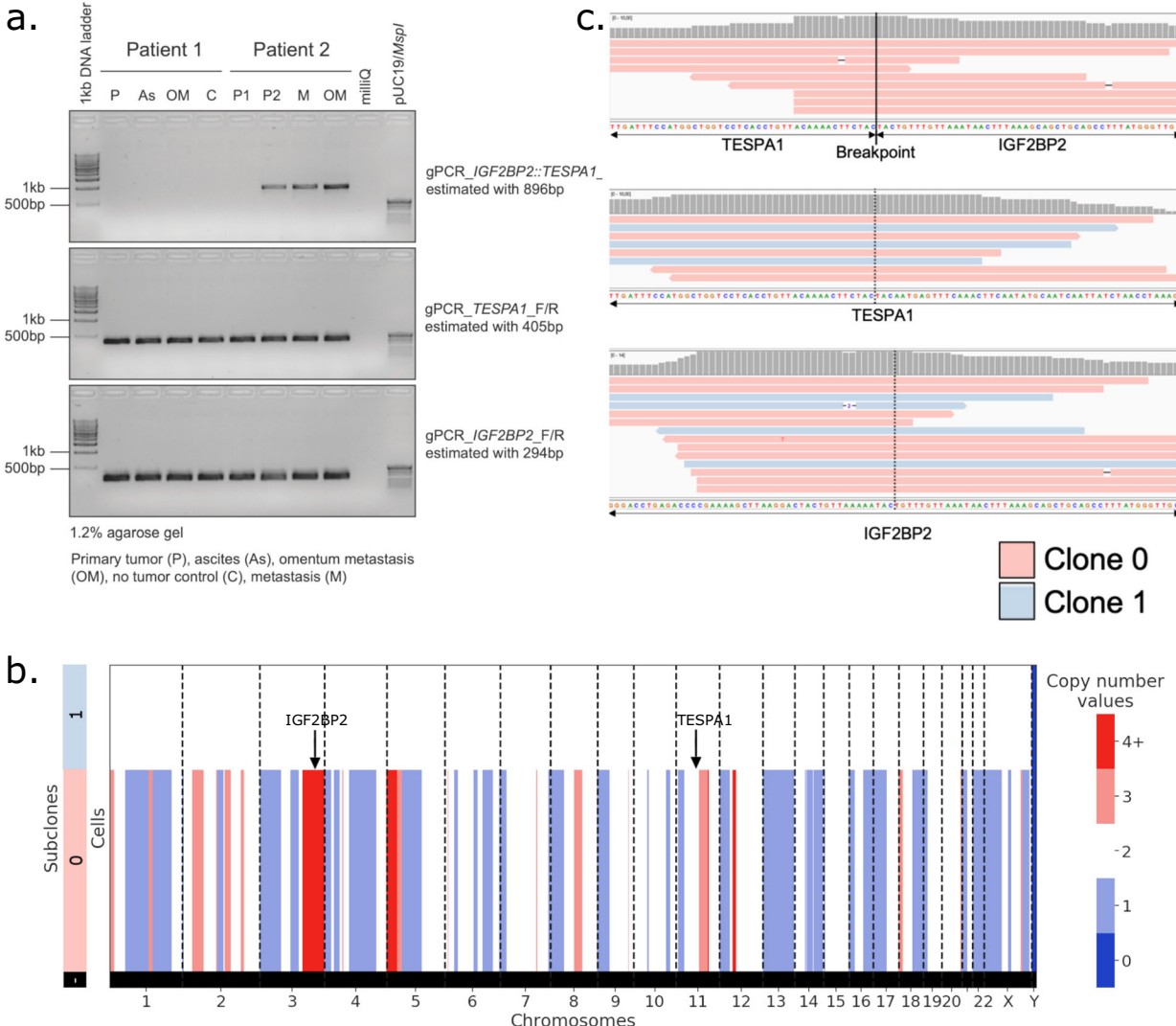

**Fig. 6 | *IGF2BP2::TESPA1* fusion breakpoint validation in bulk and scDNA.**
**a** Genotyping PCR on genomic DNA isolated from matched patient samples using gene-specific primers for *IGF2BP2::TESPA1* genomic breakpoint (top), wt *TESPA1* (middle) and wt *IGF2BP2* (bottom). *n* = 2 patients, 4 samples per patient, depending on biological material available. Source images are provided as a Source Data file. **b** Copy number values per subclone in Patient 2 scDNA-seq data. Sublone 0 has multiple copy number alterations, indicative of cancer, while Subclone 1 is copy number neutral, presumably non-cancer. **c** IGV view of scDNA reads aligning unambiguously to the *IGF2BP2::TESPA1* genomic breakpoint (top), wt *TESPA1* (middle), or wt *IGF2BP2* (bottom). In red, reads from Subclone 0 cells (cancer); in blue, reads from Subclone 1 cells (non-cancer).

Coherent with the high IGF2BP2 protein levels, *IGF2* RNA, which is bound by the wt IGF2BP2 protein, is also largely overexpressed in Patient 2 cancer cells compared to other patients (Supplementary Fig. 8d). This could indicate that the fusion happened partly due to accessible chromatin. In the ovarian cancer TCGA RNA dataset, the expression levels of exons surrounding breakpoints of *IGF2BP2* and *TESPA1* did not change (Supplementary Fig. 9a, b), and the overall expression of the genes did not correlate in any patient, suggesting that we detected an uncommon, patient-specific fusion (Supplementary Fig. 9c).

## Discussion

The detection of genomic alterations like mutations[65,66] and gene fusions[67,68], alongside isoform expression and usage[15] at the single-cell scale offers valuable insights into cancer development, metastasis formation, the role of the tumor microenvironment (TME), potential drug targets, and therapy responses[69]. Here, we applied PacBio HiFi high-throughput long-read RNA-seq on five omental metastases and matching tumor-free samples from chemo-naive HGSOC patients to detect and quantify all of these alterations.

Until now, a combination of single-cell short- and long-read sequencing was necessary to identify cell-specific isoforms: the increased depth of short-read sequencing allowed for cell typing based on gene expression, while long-read sequencing was used for isoform identification[23]. Leveraging multiple strategies to maximize the long-read sequencing output, we achieved a 50-fold increased sequencing depth compared to the first long-read PacBio scRNA-seq study[23], allowing for short-read-comparable cell-type identification. As a result, future studies employing comparable or higher long-read capacities will no longer depend on parallel short-read sequencing, leading to cost and labor savings.

Taking advantage of the long-read technology, our study provides evidence that cancer cells may induce EMT in TME mesothelial cells. In omental metastases, it has been shown that HGSOC-secreted TGF-β triggers EMT and converts TME cells, including mesothelial cells, into CAFs, which in turn may favor tumor cell adhesion, invasion, and

proliferation[70–72]. Our findings provide further evidence of this phenomenon and reveal that the process might partly be controlled through the TGF-β/miR-29/Collagen axis: secreted and endogenous TGF-β downregulates miR-29 expression, thus increasing the expression of its targets, including collagens[73]. Coincidentally, Han and colleagues[48] recently showed that omental CAF-derived exosomes from HGSOC patients contained significantly lower miR-29c than normal fibroblasts. Furthermore, the reduced levels of miR-29 have been demonstrated to play a role in the development of cisplatin resistance by upregulating collagen expression[74].

In addition to TGF-β, HGSOC cells also promote EMT and CAF activation through the secretion of other growth factors such as insulin-like growth factors (*IGF*s)[75]. In our data, HGSOC cells revealed a profoundly modified IGF system among all patients with a drastic switch from endogenous *Class I* to secreted *Class II IGF1* isoform, *IGF2* overexpression, and a highly expressed *IGF2BP2::TESPA1* gene fusion in one patient. Secreted IGF1 (Class II) and IGF2 activate EMT through IGF1R, by triggering an uncontrolled wound healing response[76]. The *IGF* gene family also promotes cancer growth, survival, proliferation, and drug resistance through signaling via *PI3K-AKT* or *MAPK*, and is a known clinical target in ovarian cancer[77].

In addition, we demonstrated the potential of the technology in terms of coverage and sequencing accuracy to detect mutations and gene fusions. In particular, in one patient, the *IGF2BP2::TESPA1* fusion was highly overexpressed compared to wt *IGF2BP2* (~10× more) and *TESPA1* (~150× more). *IGF2BP2* is known to be regulated via 3′UTR miRNA silencing[78], however, the *IGF2BP2::TESPA1* fusion has the unregulated 3′UTR of *TESPA1*, which could explain its overexpression. *TESPA1* is normally expressed in T cells[79] and long-read data confirmed T-cell-specific wt *TESPA1* expression. Short-read data however erroneously reported *TESPA1* as the most differentially expressed gene in cancer cells, resulting from 3′ end capture of the fusion transcripts. This highlights that short-read scRNA-seq data fails to distinguish between gene and fusion expression, potentially leading to wrong biological conclusions.

In addition to their canonical mRNA isoforms, numerous protein-coding genes express noncoding RNA isoforms, which are wrongly accounted for as protein-coding on the gene expression level and can serve as miRNA sponges and competing endogenous RNAs, especially in EMT and metastasis[80,81]. In our data, we found that 20% of the protein-coding gene expression was noncoding and, in the genes exhibiting a significant isoform switch between cancer and healthy cells, nearly half (49%) transitioned from a protein-coding to a noncoding isoform. Furthermore, 51% of the UMIs were composed of intrapriming and noncanonical artifacts before filtering, and their detection was only possible through isoform classification. Overall, our findings highlight the need for an isoform-specific quantification to accurately assess protein-coding gene expression[82] and narrow the RNA-protein gap[83–86]. In addition, a better isoform characterization is needed to understand their biological implications, as we partly demonstrated with *IGF1 Class I/II, cGSN, TPM2.3, RTN1-C*, and *OAS1 p42*.

Although the achieved sequencing depth allowed for short-read-independent cell typing and clustering, a further increased depth is needed to capture low-abundance transcripts. For example, we did not obtain sufficient reads to retrieve and characterize the T-cell receptor repertoire. This is consistent with a long-read scRNA-seq study in blood lymphocytes that reported a 3.6-fold lower pairing rate for T-cell receptors than the higher abundant B cell receptors from plasmablasts[87]. Using a single-cell long-read targeted sequencing approach, Byrne and colleagues were able to achieve a T-cell receptor alpha and beta pairing rate of 50% in an ovarian cancer patient sample, showing the potential to extract immune repertoire information from long-read data with increased sequencing depth[88]. Enrichment for low-abundant transcripts for

long-read sequencing or depletion of mitochondrial and ribosomal RNA[89] represent interesting avenues forward.

Taken together, we demonstrate that long-read sequencing provides a more complete picture of cancer-specific changes. These findings highlight the manifold advantages and new opportunities that this technology provides to the field of precision oncology, opening the premise of personalized drug prediction and neoantigen detection for cancer vaccines[90,91].

## Methods

### Ethics statement
The use of material for research purposes was approved by the corresponding cantonal ethic commissions Ethikkommission Nordwest- und Zentralschweiz (EKNZ: 2017–01900, to V.H.S.), and informed consent was obtained from all patients for all human primary material.

### Omentum patient cohort
Tissue samples were immediately collected from the theater and transferred on ice to the Department of Biomedicine of the University Hospital Basel for tissue dissociation.

### Sample processing
Fresh omentum and omental HGSOC tumor metastasis biopsy samples were cut into small pieces and dissociated in digestion solution (1 mg/mL collagenase/Dispase [Sigma cat. no. 10269638001], 1 unit/mL DNase I [NEB, cat. no. M0303] and 10% fetal bovine serum (FBS) in DMEM [Sigma, cat. no. D8437-500mL]) for 30 min at 37 °C. To focus on the non-adipose cell fraction, adipocytes were separated by centrifugation and the cell pellet was collected. Red blood cell lysis (RBC) was performed using MACS red blood lysis solution (cat. no. 130-094-183). Then, the cell pellet was resuspended into MACS dead cell removal microbeads (cat. no. 130-090-101) and loaded into the Auto-MACS separator to remove dead cells. After counting cell numbers, cells were resuspended in PBS with 1% BSA and transferred to the Genomics Facility Basel. The cell suspension was again filtered and cell number and viability were assessed on a Cellometer K2 Image Cytometer (Nexcelom Bioscience, cat. no. Cellometer K2) using ViaStain AOPI Staining Solution (Nexcelom Bioscience, cat. no. CS2-0106-5mL) and PD100 cell counting slides (Nexcelom Bioscience, cat. no. CHT4-PD100-003). For samples with viability below 70% and when cell numbers allowed (>10$^5$ cells total), apoptotic and dead cells were removed by immunomagnetic cell separation using the Annexin Dead Cell Removal Kit (StemCell Technologies, cat. no. 17899) and EasySep Magnet (StemCell Technologies, cat. no. 18000). If the cell pellet appeared still red, additional RBC lysis was performed. Cells were washed with a resuspension buffer (PBS with 0.04% BSA), spun down, and resuspended in a resuspension buffer. Finally, cells were again counted and their viability was determined. The cell concentration was set according to 10x Genomics protocols (700–1200 cells/μL).

### 10x Genomics single-cell capture and short-read sequencing
Cell suspensions were loaded and processed using the 10x Genomics Chromium platform with the 3P v3.1 kit on the 10x Genomics Chromium Single Cell Controller (10x Genomics, cat. no. PN-120263) according to the manufacturer's instructions. In total, 500 or 1000 cells were targeted per lane. The quality of cDNA traces and GEX libraries were profiled on a 5200 Fragment Analyzer (Agilent Technologies).

Paired-end sequencing was performed on the Illumina NovaSeq platform (100 cycles, 380 pm loading concentration with 1% addition of PhiX) at recommended sequencing depth (20,000–50,000 reads/cell).

**Long-read library preparation and PacBio sequencing.** To increase long-read PacBio sequencing throughput, we followed the strategy of

cDNA concatenation of the HIT-scISOseq protocol[24] with the modification of two rounds of biotin-PCR in order to further reduce template-switch oligo (TSO) artifacts from the data.

**cDNA amplification and biotin enrichment.** In total, 15 ng of each patient's cDNA library were amplified using the KAPA HiFi Hot-Start Uracil+ ReadyMix 2x (Kapa Biosystems, cat. no. KK2801) with 0.5 μM final concentration of custom-primers (Integrated DNA Technologies, HPLC purified). Primers contained overhang sequences adapted from ref. 92 with a single deocxyuredine (dU) residue at a 10 nt distance from the 5′ terminus enabling USER enzyme digestion and creating single-stranded overhangs. Generated PCR fragments thus contain a single dU residue per DNA strand. The forward primer was specific to the 10x Genomics partial Read 1 sequence and contained a biotin modification allowing for biotin enrichment of amplified full-length cDNA molecules. The reverse primer was specific to the 10x Genomics partial TSO sequence.

Forward primer: /5Biosg/AGGTCTTAA/ideoxyU/CTACACGACGCCTTCCGATCT

Reverse primer: ATTAAGACC/ideoxyU/AAGCAGTGGTATCAACGCAGAG.

The PCR was run according to the manufacturer's instruction with two cycles at an annealing temperature of 63 °C followed by seven cycles at an annealing temperature of 67 °C; the annealing time was 30 s. Extension was performed at 72 °C for 90 s. PCR products were purified at 0.6× SPRIselect bead cleanup (Beckman Coulter, cat. no. B23318) according to the manufacturer's instructions and eluted in 22 μL EB buffer (Qiagen, cat. no. 19086). DNA concentrations were measured using the Qubit dsDNA HS Assay Kit (ThermoFisher Scientific, cat. no. Q32854), which were in the range of 1.5 μg per sample. cDNA traces were additionally evaluated on a 5200 Fragment Analyzer System (Agilent Technologies) using the HS NGS Fragment Kit, 1–6000 bp (Agilent, cat. no. DNF-474-0500). Full-length cDNAs were enriched through capture on 5 μL streptavidin-coated M-280 dynabeads using the Dynabeads™ kilobaseBINDER™ Kit (Invitrogen, cat. no. 60101), thus depleting TSO-TSO artifacts. Washed Dynabeads containing the DNA-complexes were directly resuspended in 20 μL USER reaction buffer containing 10 μL StickTogether DNA Ligase Buffer 2x (NEB, cat. no. B0535S), 1.5 μL USER Enzyme (NEB, cat. no. M5505S) and 8.5 μL Nuclease-free water (Invitrogen, AM9939) and incubated in a thermocycler at 37 °C for 20 min and held at 10 °C (no annealing). This created a nick at the deoxyuracil site forming palindrome overhangs and releasing the biotin-bound DNA molecules from the beads. Beads were removed by magnetic separation and the supernatant with the biotin-released cleaved PCR products was subjected to a 0.6× SPRIselect cleanup step. Approximately 100 ng of purified product per sample were split into two aliquots and subjected to a second PCR amplification step with six cycles using an annealing temperature of 67 °C. Reactions were pooled, purified by 0.6× SPRIselect cleanup, and quality checked on both Qubit and Fragment Analyzer. Total DNA yield was between 5 and 8 μg, which were subjected to a second round of streptavidin-purification using 10 μL of beads.

**Transcript ligation.** Beads were incubated in 19 μL USER reaction buffer at 37 °C for 20 min for USER digestion and 25 °C for 17 min for overhang annealing. Beads were then removed by magnetic separation and the supernatant was transferred to a new PCR tube. In total, 1 μL of T4 DNA ligase high-concentration (2,000,000, units/mL, NEB, cat. no. M0202T) was added, mixed, and incubated at 10 °C for >24 h and heat-inactivated at 65 °C for 10 min. To efficiently deplete any non-ligated transcripts, 0.38× SPRIselect cleanup was performed, eluted in 20 μL EB buffer and traces were evaluated on the Fragment Analyzer using the HS Large Fragment kit (Agilent Technologies, cat. no. DNF-492-0500) at 1:5 dilutions. Ligation products were 8–11 kb long; average yield was 100 ng per sample.

**End repair/dA tailing, adapter ligation, and PCR amplification.** To enable PCR amplification of the ligated construct, the NEBNext Ultra II DNA Library Prep Kit for Illumina was followed (NEB, cat. no. E7645S) using total DNA yield as input material. In total, 2.5 μL of 5 μM dT overhang adapter (Roche, cat. no. KK8727) was used for the End Prep reaction. Adapter-ligated libraries were purified by 0.39× SPRIselect cleanup, eluted in 22 μL EB buffer and products were evaluated by HS Large Fragment kit. Total yield of around 40 ng was split in two and PCR amplified using 2× KAPA HiFi Hot-Start ReadyMix (Roche, cat. no. KK2602) and KAPA Library Amplification Primer Mix (10× concentration, Roche, cat. no. KK2623), 10 μL library input each with 11 cycles and 9 min extension time. Following a 0.38× SPRIselect cleanup and elution in 48 μL EB buffer, products were evaluated on a large fragment gel revealing an average fragment length of libraries of 4.6 kb and an average total of 1.1 μg DNA. To increase the total yield to 2 μg DNA required for SMRTbell library preparation of a product with 5 kb amplicon size, the PCR was repeated with three additional cycles and 5 min extension time. After 0.4× SPRI cleanup and Fragment Analyzer inspection, the final yield was 2 μg per library.

**PacBio SMRTbell library preparation.** The SMRTbell Express Template Kit (PacBio, cat. no. 100-938-900) was used following the manufacturer's instructions for DNA damage repair, end repair/dA tailing and ligation of a hairpin adapter (double amount used). Final purification of the SMRTbell template was performed by 0.42× SPRIselect cleanup and elution in 43 μL EB buffer. Exonuclease treatment was performed by the addition of 5 μL of NEBbuffer1 (NEB, cat. no. B7001S) and 1 μL of each Exonuclease I (NEB, cat. no. M0293S) and Exonuclease III (NEB, cat. no. M0206S) bringing the total volume to 50 μL per reaction. Enzyme treatment was performed at 37 °C for 60 min. After SPRIselect cleanup, products were quantified on a large fragment gel at 1:30 dilution. The final yield was approximately 650 ng per sample, a sufficient amount for long-read sequencing.

**PacBio Sequel II sequencing.** Libraries were sequenced on the PacBio Sequel II platform with the SMRT cell 8M. Omentum metastasis and tumor-free omentum were run on three and two 8M cells, respectively.

### Genotyping PCR on genomic DNA

Genomic DNA was extracted from homogenized tumor tissue samples (*n* = 8 samples matching sampling time, Basel Ovarian Biobank) derived from patients using the DNeasy Blood & Tissue kit (QIAGEN, cat. no. 69504). Isolated DNA underwent QC using Nanodrop and Qubit measurements. Genotyping PCR on genomic DNA was performed using the MyTaq DNA Polymerase system (Bioline, cat. no. BIO-21105). Briefly, MyTaq reaction buffer and MyTaq DNA polymerase were pipetted together with 200 nM primer (Sigma Aldrich) pairs (gPCR_IGF2BP2-TESPA1_Bp_F 5′-CCT GCT TTG AGG AGG GGA GGG A-3′ & gPCR_IGF2BP2-TESPA1_Bp_R 5′-ACT GAG GAC AAT GCT ACG CAA GA-3′; gPCR_TESPA1_F 5′-CCT GCT TTG AGG AGG GGA GGG A-3′ & gPCR_TESPA1_R 5′-TGA GAA CTG CTG TTC CAG GAG ACA-3′; gPCR_IGF2BP2_F 5′-ACA CTG GAC CCA TGC TTG AGC T-3′ & gPCR_IGF2BP2_R 5′-GCG TGC TAT GAA CAC TCC AGG CC-3′), and 50 ng genomic DNA (gDNA). PCR conditions were 1 cycle at 94 °C for 5 min followed by 35 cycles at 95 °C for 20 s, 58 °C for 15 s, 72 °C for 1 min, and finished with 1 cycle at 72 °C for 5 min. Amplicons were visualized on a 1.2% agarose gel together with DNA ladder.

### Immunofluorescence

Formalin-fixed and paraffin-embedded tissue samples were obtained from the Basel Ovarian Biobank matching with patients 1 and 2 on sampling time and site. Briefly, samples were deparaffinized and immersed for 10 min in a 10 mM sodium citrate buffer at pH 6.0 (Sigma Aldrich, cat. no. C9999) at 95 °C for antigen retrieval. Samples were permeabilized in 0.25% (v/v) Triton™ X-100 (Roth, cat. no. 3051.3) in

PBS (Sigma Aldrich, cat. no. D8537-500M) for 5 min and blocked in 5% FBS (Sigma Aldrich, cat. no. F7524-500ml), 0.1% Triton™ X-100, 1% BSA (Roth, Fraction V, cat. no. 8076.4) in PBS for 1 hour. The following antibodies were used for this study: rabbit IGF2BP2 (C-terminal-specific, ThermoFisher Scientific, cat. no. MA5-42874), EpCAM (Cell Signaling Technologies, cat. no. 5488S,) and goat anti-rabbit Alexa Fluor® 647 (Cell Signaling Technology, cat. no. 4414). Slides were mounted using ProLong® Gold Antifade Reagent with DAPI (Cell Signaling Technology, cat. no. 8961). Images were acquired using a Nikon spinning-disk confocal microscope (Nikon CSU-W1 spinning-disk confocal microscope) and processed with Fiji. Cell quantification was performed using an in-house developed QuPath script for cell detection and annotations.

### Single-cell DNA sequencing
Cell suspensions were loaded and processed using the 10x Genomics Chromium platform with the single-cell CNV kit on the 10x Genomics Chromium Single Cell Controller (10x Genomics, cat. no. PN-120263) according to the manufacturer's instructions. Paired-end sequencing was performed on the Illumina NovaSeq platform (100 cycles, 380 pm loading concentration with 1% addition of PhiX) at recommended sequencing depth.

### Data analysis
**Statistics and reproducibility.** This study is a pilot study to demonstrate the potential of single-cell long-read sequencing in oncology. The number of patients was limited to three in order to secure sufficient sequencing depth. No statistical method was used to predetermine sample size. For comparisons between two groups (fold-change expression between miR-29-targeted and non-targeted genes), the two-tailed Student's t-test was used. For cell-type or clonal expression comparison, the two-tailed Fisher's exact test was used. No data were excluded from the analyses. The experiments were not randomized. The Investigators were not blinded to allocation during experiments and outcome assessment.

### Short-read data analysis
**Preprocessing.** Raw reads were mapped to the GRCh38 reference genome using 10x Genomics Cell Ranger 3.1.0 to infer read counts per gene per cell. We performed index-hopping removal using a method developed by Griffiths et al.[93].

**10x Genomics read data processing, normalization, and visualization.** Expression data of each sample was analyzed using the scAmpi (v1.0) workflow[94]. In brief, UMI counts were quality-controlled and cells and genes were filtered to remove mitochondrial and ribosomal contaminants. Cells for which over 50% of the reads mapped to mitochondrial genes and cells with fewer than 400 genes expressed were removed. By default, all non-protein-coding genes, genes coding for ribosomal and mitochondrial proteins, and genes that were expressed in less than 20 cells were removed. Doublet detection was performed using scDblFinder[95]. Subsequently, counts were normalized with sctransform[96], regressing out cell cycle effects, library size, and sample effects as nonregularized dependent variables. Similar cells were grouped based on unsupervised clustering using Phenograph[97], and automated cell-type classification was performed independently for each cell[98] using gene lists defining highly expressed genes in different cell types. Major cell-type marker lists were developed in-house based on unpublished datasets (manuscripts in preparation), including the Tumor Profiler Study[99], using the Seurat FindMarkers method[100]. Immune subtype marker gene lists were obtained from ref. 101 and enriched with T-cell subtypes from ref. 102 The results of the unsupervised clustering and cell typing are visualized in a low-dimensional representation using Uniform Manifold Approximation and Projection (UMAP).

### Long-read data analysis
**Generating CCS.** Using SMRT-Link (v9.0.0.92188), we performed circular consensus sequencing (CCS) with the following modified parameters: maximum subread length 50,000 bp, minimum subread length 10 bp, and minimum number of passes 3.

**Unconcatenating long reads.** We used NCBI BLAST (v2.5.0+) to map the 5' and 3' primers to CCS constructs, with parameters: "-outfmt 7 -word_size 5" as described previously[24]. Sequences between two successive primers were used as input for primer trimming using IsoSeq3 (v3.4) Lima (parameters: −isoseq −dump-clips −min-passes 3). Cell barcodes and UMIs were then demultiplexed using IsoSeq3 tag with parameter −design T-12U-16B. Finally, we used IsoSeq3 refine with option −require-polya to remove concatemers and trim polyA tails. Only reads with a correct 5'−3' primer pair, a barcode also found in the short-read data, a UMI, and a polyA tail were retained.

**Isoform classification.** Demultiplexing UMIs with IsoSeq3 dedup and calling isoforms on the cohort level with collapse_isoforms_by_sam.py resulted in unfeasible runtimes. Therefore, we called isoforms first on the cell level as a pre-filtering step. Long reads were split according to their cell barcodes, and UMI deduplication was performed using IsoSeq3 dedup. Next, reads were mapped and aligned to the reference genome (hg38) using minimap2 (v2.17) with parameters: -ax splice -uf −secondary=no -C5. Identical isoforms were merged based on their aligned exonic structure using collapse_isoforms_by_sam.py with parameters: -c 0.99 -i 0.95 −gen_mol_count. We then classified isoforms using SQANTI3[32] (v1.6) with arguments: −skipORF −fl_count −skip_report. We finally filtered artifacts including intrapriming (accidental priming of pre-mRNA 'A's), reverse-transcriptase template-switching artifacts, and mismapping to noncanonical junctions. To create a unique isoform catalog for all our samples, we then retained only reads associated with isoforms passing the SQANTI3 filter, and we ran collapse_isoforms_by_sam.py, SQANTI3 classification and filtering again on all cells together. The described pipeline is available here and was implemented in Snakemake, a reproducible and scalable workflow management system[103].

**3' and 5' isoform filtering.** For SQANTI3-defined isoforms, "incomplete splice match", "novel in catalog", and "novel not in catalog", we only retained isoforms falling within 50 bp of a CAGE-validated transcription start site (FANTOM5 CAGE database), and 50 bp of a polyA site form the PolyASite database[34]. All "full splice match" isoforms were retained. The GENCODE database was used as a comparison, all protein-coding isoforms were grouped under the GENCODE.full label, a subset including only full-length isoforms was labeled as GENCOD.FL, and the Matched Annotation from NCBI and EMBL-EBI (MANE[35]) was named GENCODE.MANE.

**Cell-type-specific isoforms.** Considering only the SQANTI3-defined "full splice match", "novel not in catalog", and "novel in catalog" isoforms with at least three reads, we established the following classification: "Cell-specific" isoforms are present in only 1 cell and "cell type-specific" isoforms are present in >=3 cells of a unique cell type.

**Cell-type annotation.** Cells were annotated with long reads the same way as short reads, using scROSHI. The tissue cell types were modified according to gene expression in long reads. Immune subtype marker gene lists were unchanged.

**Normalization and visualization.** Long-read gene expression counts were normalized and visualized as described above for short reads. Long-read gene expression counts were normalized using 10,000 features instead of the default 3000 in sctransform[96].

**Mutation detection.** Positions of mutations from Foundation Medicine's targeted NGS panel (Foundation One CDx) mutations described in Supplementary Data 1 were used as references. One mutation not present in the list, TP53_P151H, was detected in IGV in Patient 1 and added to the list. If a position was mutated at least in one cell belonging to a distal biopsy sample, the mutation was classified as a germline variant. Cells with one mutated read in one of the positions were considered mutated.

**Alternative polyadenylation (APA) analysis.** To analyze differences between 3'UTR lengths, we used a modified version of DaPars2[104] (https://github.com/ArthurDondi/DaPars2_LR), with an APA site detection adapted to long-read coverage.

Briefly, we identified the 3'UTR exon of each isoform, and overlapping 3'UTR exons with different 5' start positions were discarded from analysis, as they create false positive APAs. Then, for each remaining 3'UTR, we computed the coverage for each cell type. The distal site position was defined as the most 3' position with coverage superior to 10 in any cell type:

$$L^* = \max\left(\{k : w_k^c > 10, 1 < k < L, 1 < c < m\}\right) \quad (1)$$

where $L$ is the length of the annotated 3' UTR region and $L^*$ is the defined distal site position. $w_k^c$ is the coverage in cell type $c$ at position $k$, and $m$ is the total number of cell types.

We inferred the exact location of APA sites by maximizing the coverage gap between the 50 positions before and after the possible APA sites, based on the two-polyA-site model, the most common model of APA regulation:

$$(C^*, P^*) = argmax_{1 < c < m, 1 < P < L^*}\left(\frac{1}{50}\sum_{i=1}^{50} w_{P+i}^c - \frac{1}{50}\sum_{i=1}^{50} w_{P-i}^c\right)^2 \quad (2)$$

where $P$ is the estimated length of alternative proximal 3' UTR, and the optimal proximal site $P^*$ in cell type $C^*$ is the one with the maximal objective function value. $w_k^c$ is the coverage at the position $k$ and cell type $c$, and $m$ is the total number of cell types.

Then distal and proximal site coverages $W_d$ and $W_p$ in condition $C$ were defined as:

$$W_d = \frac{1}{50}\sum_{i=1}^{50} w_{L^*-i}^c, \quad W_p = \frac{1}{50}\sum_{i=1}^{50} w_{P^*-i}^c \quad (3)$$

The fraction of distal polyA site usage is then defined as:

$$F = \frac{w_d}{w_d + w_p} \quad (4)$$

The degree of difference in APA usage in cell types $C_1$ and $C_2$ can be quantified as a *Fraction Change*, which is capable of identifying 3' UTR lengthening (positive index) or shortening (negative index):

$$Fraction\,Change = F_{C_1} - F_{C_2} \quad (5)$$

$P$ values from a two-sided Fisher's exact test, comparing distal and proximal site coverages $W_d$ and $W_p$ between cell types $C_1$ and $C_2$, were reported per APA site. P values across all APA site were then corrected using the Benjamini–Hochberg (BH) correction for multiple testing with a false discovery rate of 5%.

**Differential isoform tests.** Differential isoform testing was performed using a $\chi^2$ test as previously described in Scisorseq[26] (v.0.1.2). Briefly, counts for each isoform ID were assigned to individual cell types, and genes were discarded if they were mitochondrial, ribosomal, or if they

did not reach sufficient depth per condition (25 reads per condition per gene). $P$ values from a $\chi^2$ test for differential isoform usage were computed per gene where a sufficient depth was reached, and we corrected for multiple testing using Benjamini–Hochberg correction with a 5% false discovery rate. If the corrected $P$ value was ≤0.05 and the sum of change in the relative percent of isoform (ΔΠ) of the top two isoforms in either positive or negative direction was more than 20%, then the gene was called differentially spliced. To classify the top differentially spliced genes, we took the rank of genes by ΔΠ and corrected p values, and took the square root of the multiplication of those two ranks. The smallest ranks obtained this way were considered as the top differentially expressed genes. Differentially used isoforms were visualized using ScisorWiz[105].

**Isoform biotypes.** Biotypes of novel isoforms were assessed internally by the GENCODE team with biotypes matching those described in ref. 106. Known isoform biotypes were retrieved from the GENCODE v36 database.

**Biotype change analysis.** In genes with differential isoform usage between conditions, we compared the biotypes of the most expressed isoform of each condition, and if they were not identical this was considered a change in biotype.

**Protein-coding gene expression.** Considering only genes with protein-coding biotype (GENCODE) with a minimal expression of 20 UMIs in a condition, we assessed the ratio of noncoding isoforms (NMD, intron retention, etc.) expression to total gene expression (noncoding and coding).

**Pathway enrichment analysis.** We used GSVA to perform pathway enrichment analysis. When comparing TME and distal cells, gene sets were obtained from MSigDB[107], except for the miR-29 targets that were obtained from Cushing and colleagues[108]. When we searched for pathways enriched in lengthened or shortened 3'UTRs in cancer cells, we used the investigate function of the GSEA webpage (https://www.gsea-msigdb.org/gsea/msigdb/human/annotate.jsp), and compared against all MSigDB gene sets[107].

**Fusion discovery.** Mapped reads from isoform classification were pooled. We called reads mapping to two separate genes at a distance of more than 100,000 bp or to different chromosomes using fusion_finder.py (cDNA_Cupcake package, https://github.com/Magdoll/cDNA_Cupcake/releases/tag/v28.0.0) with parameters –min_locus_coverage_bp 200 -d 1000000. Fusion isoforms with sufficient depth (min. 10 reads) were kept, and their breakpoint, expression per cell type, and number of cells in which they are expressed were assessed.

**Short-reads re-alignment to *IGF2BP2::TESPA1*.** We designed a custom reference including *IGF2BP2::TESPA1* transcriptomic breakpoint as well as the wild-type *IGF2BP2* and *TESPA1* exon junction covering the breakpoint. The reference was composed of 6 sequences of 80 nucleotides (40 bases upstream and downstream of the breakpoint), sequences XXX_1 and XXX_2 represent the breakpoints of the two main isoforms seen in each gene:

>TESPA1_wt_1
TTCTGTCAGACCACATGCTGTTGTGGTGGTGGAGAAAGCAATTCTGGAGGCTGGCAAATCCAAGGTCAAAAGCCTGCA
>TESPA1_wt_2
TTCTGTCAGACCACATGCTGTTGTGGTGGTGGAGAAAGCTTCACGAGTCTTTGCCAGCAAAAGTCTGGTGGTGGTGGG
>IGF2BP2_wt_1

ATGTGACGTTGA-
CAACGGCGGTTTCTGTGTCTGTGTTGACTTGTTCCACATTCTCCACTG
TCCCATATTGAGCCAAAA

>IGF2BP2_wt_2
ATCACTGGATTGTGTGTTCTTCTGAATTACTTCTTTAGGCTTGT
TCCACATTCTCCACTGTCCCATATTGAGCCAAAA

>TESPA1_IGF2BP2_fusion_1
TTCTGTCAGACCACATGCTGTTGTGGTGGTGGAGAAAGCCTTG
TTCCACATTCTCCACTGTCCCATATTGAGCCAAAA

>TESPA1_IGF2BP2_fusion_2
CAAATCCAAGGTCAAAAGCCTGCATCGGTGAGGGCCTCC
TTGTTCCACATTCTCCACTGTCCCATATTGAGCCAAAA

Patient 2 reads were aligned to this reference using minimap2 with parameters: -ax sr –secondary=no. Reads mapping unambiguously to one of those reference sequences were then attributed to the cell type to which their cell barcode belonged.

### scDNA analysis

Cell Ranger DNA was used to demultiplex and align Chromium-prepared sequencing samples. We used the cellranger-dna (v1.1.0) mkfastq command to generate FASTQ files from the Illumina raw BCL files, and we ran the command cellranger-dna cnv to align FASTQ files to the hg38 reference genome, call cells, and estimate copy numbers. We obtained the copy number profiles and detected the main clonal structure of samples using SCICoNE[109].

### DNA breakpoint validation

To validate in scDNA data breakpoints found in scRNA data, we used the putative scRNA breakpoint reads as a reference to re-align scDNA reads using BWA (v0.7.17) with options: -pt8 -CH. For the *IGF2BP2::TESPA1* fusion, the reference was composed of three sequences of 184 nucleotides (92 bases upstream and downstream of the breakpoint):

>IGF2BP2_WT
CAAACTTGTAGAAATGTGAATTTTTCTTGTTATTTTACAAGATTT
GCAAAGGGACCTGAGACCCCGAAAAGCTTAAGGACTACTGTTAAAAA
TACTGTTTGTTAAATAACTTTAAAGCAGCTGCAGCCTTTATGGGTTG
CAGGGAGTTGTATGTAATGCTCAGAAAGAGCTGCCACTGAGAAT

>TESPA1_WT
TTCAATGATGTGGGCTGATTAGAACATAGCTGAAAGCAGGTGTT
GGGATATTGATTTCCATGGCTGGTCCTCACCTGTTACAAAACTTC
TACTACAATGAGTTTCAAACTTCAATATGCAATCAATTATCTAAC
CTAAAGATCTTGGTAAAACTGTGATTCATTAGGTCTGGGGTEGAGCTG

>IGF2BP2_TESPA1_Fusion
TTCAATGATGTGGGCTGATTAGAACATAGCTGAAAGCAGGTG
TTGGGATATTGATTTCCATGGCTGGTCCTCACCTGTTACAAAACTTC
TACTACTGTTTGTTAAATAACTTTAAAGCAGCTGCAGCCTTTATGGGT
TGCAGGGAGTTGTATGTAATGCTCAGAAAGAGCTGCCACTGAGAAT

Reads mapping unambiguously to one of those reference sequences were then attributed to the clone to which their cell barcode belonged.

### Reporting summary

Further information on research design is available in the Nature Portfolio Reporting Summary linked to this article.

## Data availability

The raw sequencing files, as well as the associated analysis files (isoforms gff file, reads associated to isoforms, and the Gencode annotation of novel isoforms) reported in this study have been deposited in the European Genome-phenome Archive (EGA) under the accession number EGAS00001006807. Files are available indefinitely for non-commercial and not-for-profit use only, under restricted access in compliance with data privacy laws, via inquiry to the Data Access Committee of the Faculty of Medicine at the University of Basel <med-dac@unibas.ch>. We aim to respond to all initial requests under 2 weeks, and data access should happen within two months. Patient 2 scDNA reads mapping the *IGF2BP2::TESPA1* fusion breakpoint are available at https://eth-nexus.github.io/tu-pro_website/publications/dondi_et_al_2022/. Source data images are provided at https://doi.org/10.5281/zenodo.10036378[110]. The TCGA data was obtained at https://portal.gdc.cancer.gov/repository, with the following specifications: Project=TCGA-OV, DataType=Gene Expression Quantification. Human genome hg38 is available at https://hgdownload.soe.ucsc.edu/goldenPath/hg38/bigZips/hg38.fa.gz. Gencode v36 gene annotation used in this study is available at https://ftp.ebi.ac.uk/pub/databases/gencode/Gencode_human/release_36/gencode.v36.annotation.gtf.gz. All additional information will be made available upon reasonable request to the authors.

## Code availability

The code used to pre-process, analyze the data, and generate the figures of this study has been deposited in the GitHub repository: https://github.com/cbg-ethz/scIsoPrep[111]. The code to analyze differential polyadenylation site in long reads has been deposited in the GitHub repository: https://github.com/ArthurDondi/DaPars2_LR[112].

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

## Acknowledgements

We thank Ina Nissen (Genomics Facility Basel) for technical support with PacBio sequencing, Ching-Yeu Liang (University Hospital Basel) for assistance with sample dissociation, and Anne Bertolini for her help with the scAmpi R package. We also wish to thank Elisabeth Tseng (Pacific Bioscience) for her help with cDNA_Cupcake, SQANTI3, and long-read sequencing analysis, Anoushka Joglekar (Weill Cornell Medicine) for her help with ScisorSeq and ScisorWiz as well as Lucia Csepregi for her help with immune repertoire analysis. We thank Olivier Belli for his help with 3'UTR shortening mechanisms. Furthermore, we are grateful to Adam Frankish (Wellcome Sanger Institute) and the GENCODE Project team for their help with the annotation of isoforms. Graphical illustrations were created with BioRender.com. Illumina sequencing was carried out in the Genomics Facility Basel of the University of Basel and the Department of Biosystems Science and Engineering, ETH Zurich. PacBio sequencing was done in the Functional Genomics Center Zurich of the University of Zurich and ETH Zurich, and the Lausanne Genomic Technologies Facility, University of Lausanne. This work was supported by the SNSF SPARK grant (#190413 to C.B. and U.L.), the Strategic Focal Area "Personalized Health and Related Technologies" grants of the ETH Domain (PHRT #2017-510 to C.B. and PHRT Pioneer Project #715 to V.H.S. and F.J.). N. Borgsmüller and A.D. were supported by the European Union's Horizon 2020 research and innovation program under the Marie Sklodowska-Curie grant agreement (#766030 to N. Beerenwinkel).

## Author contributions

U.L. and C.B. acquired funding and conceived and designed the experiments. V.H.S. provided patient material. U.L., F.J., A.D., and C.B. selected the clinical cohort. U.L. performed 10x Genomics sample processing as well as short-read and long-read sequencing library preparation. The Tumor Profiler Consortium provided scDNA-seq data and NGS panel results. A.D. designed the preprocessing pipeline and implemented it with the help of N. Borgsmüller. A.D. conducted all computational analyses. F.J. and R.C. performed experimental validation. F.S. assisted in short-read scRNA-seq analysis. A.D., U.L., F.J., N. Beerenwinkel, and C.B. interpreted the data. A.D. and U.L. wrote the manuscript with contributions from all authors. All authors read and approved the final manuscript.

## Competing interests

The authors declare no competing interests.

## Additional information

## Tumor Profiler Consortium

Rudolf Aebersold[5], Melike Ak[6], Faisal S. Al-Quaddoomi[2,4], Silvana I. Albert[7], Jonas Albinus[7], Ilaria Alborelli[8], Sonali Andani[2,9,10,11], Per-Olof Attinger[12], Marina Bacac[13], Daniel Baumhoer[8], Beatrice Beck-Schimmer[14], Christian Beisel[1 ✉], Christian Beisel[4], Lara Bernasconi[15], Anne Bertolini[2,4], Bernd Bodenmiller[16,17], Ximena Bonilla[2,9,10], Lars Bosshard[2,4], Byron Calgua[8], Ruben Casanova[17], Stéphane Chevrier[17], Natalia Chicherova[2,4], Ricardo Coelho[3], Maya D'Costa[12], Esther Danenberg[18], Natalie Davidson[2,9,10], Monica-Andreea Drăgan[4], Reinhard Dummer[6], Stefanie Engler[17], Martin Erkens[19], Katja Eschbach[4], Cinzia Esposito[18], André Fedier[3], Pedro Ferreira[4], Joanna Ficek[2,9,10], Anja L. Frei[11], Bruno Frey[20], Sandra Goetze[7], Linda Grob[2,4], Gabriele Gut[18], Detlef Günther[21], Martina Haberecker[11], Pirmin Haeuptle[22], Viola Heinzelmann-Schwarz[3,23], Sylvia Herter[13], Rene Holtackers[18], Tamara Huesser[13], Alexander Immer[9,24], Anja Irmisch[6], Francis Jacob[3], Andrea Jacobs[17], Tim M. Jaeger[12], Katharina Jahn[4], Alva R. James[2,9,10], Philip M. Jermann[8], André Kahles[2,9,10], Abdullah Kahraman[2,11], Viktor H. Koelzer[11], Werner Kuebler[25], Jack Kuipers[1,2], Christian P. Kunze[26], Christian Kurzeder[27], Kjong-Van Lehmann[2,9,10],

Mitchell Levesque[6], Ulrike Lischetti[3], Sebastian Lugert[12], Gerd Maass[20], Markus G. Manz[28], Philipp Markolin[2,9,10], Martin Mehnert[7], Julien Mena[5], Julian M. Metzler[29], Nicola Miglino[22], Emanuela S. Milani[7], Holger Moch[11], Simone Muenst[8], Riccardo Murri[30], Charlotte K. Y. Ng[8,31], Stefan Nicolet[8], Marta Nowak[11], Monica Nunez Lopez[3], Patrick G. A. Pedrioli[32], Lucas Pelkmans[18], Salvatore Piscuoglio[3,8], Michael Prummer[2,4], Natalie Rimmer[3], Mathilde Ritter[3], Christian Rommel[19], María L. Rosano-González[2,4], Gunnar Rätsch[2,9,10,32], Natascha Santacroce[4], Jacobo Sarabia del Castillo[18], Ramona Schlenker[33], Petra C. Schwalie[19], Severin Schwan[12], Tobias Schär[4], Gabriela Senti[15], Wenguang Shao[7], Franziska Singer ⑮ [2,4], Sujana Sivapatham[17], Berend Snijder[2,5], Bettina Sobottka[11], Vipin T. Sreedharan[2,4], Stefan Stark[2,9,10], Daniel J. Stekhoven[2,4], Tanmay Tanna[4,9], Alexandre P. A. Theocharides[28], Tinu M. Thomas[2,9,10], Markus Tolnay[8], Vinko Tosevski[13], Nora C. Toussaint[2,4], Mustafa A. Tuncel[1,2], Marina Tusup[6], Audrey Van Drogen[7], Marcus Vetter[34], Tatjana Vlajnic[8], Sandra Weber[15], Walter P. Weber[35], Rebekka Wegmann[5], Michael Weller[36], Fabian Wendt[7], Norbert Wey[11], Andreas Wicki[28,37], Mattheus H. E. Wildschut[5,28], Bernd Wollscheid[7], Shuqing Yu[2,4], Johanna Ziegler[6], Marc Zimmermann[2,9,10], Martin Zoche[11] & Gregor Zuend[38]

[5]ETH Zurich, Department of Biology, Institute of Molecular Systems Biology, Otto-Stern-Weg 3, 8093 Zurich, Switzerland. [6]University Hospital Zurich, Department of Dermatology, Gloriastrasse 31, 8091 Zurich, Switzerland. [7]ETH Zurich, Department of Health Sciences and Technology, Otto-Stern-Weg 3, 8093 Zurich, Switzerland. [8]University Hospital Basel, Institute of Medical Genetics and Pathology, Schönbeinstrasse 40, 4031 Basel, Switzerland. [9]ETH Zurich, Department of Computer Science, Institute of Machine Learning, Universitätstrasse 6, 8092 Zurich, Switzerland. [10]University Hospital Zurich, Biomedical Informatics, Schmelzbergstrasse 26, 8006 Zurich, Switzerland. [11]University Hospital Zurich, Department of Pathology and Molecular Pathology, Schmelzbergstrasse 12, 8091 Zurich, Switzerland. [12]F. Hoffmann-La Roche Ltd, Grenzacherstrasse 124, 4070 Basel, Switzerland. [13]Roche Pharmaceutical Research and Early Development, Roche Innovation Center Zurich, Wagistrasse 10, 8952 Schlieren, Switzerland. [14]University of Zurich, VP Medicine, Künstlergasse 15, 8001 Zurich, Switzerland. [15]University Hospital Zurich, Clinical Trials Center, Rämistrasse 100, 8091 Zurich, Switzerland. [16]ETH Zurich, Institute of Molecular Health Sciences, Otto-Stern-Weg 7, 8093 Zurich, Switzerland. [17]University of Zurich, Department of Quantitative Biomedicine, Winterthurerstrasse 190, 8057 Zurich, Switzerland. [18]University of Zurich, Institute of Molecular Life Sciences, Winterthurerstrasse 190, 8057 Zurich, Switzerland. [19]Roche Pharmaceutical Research and Early Development, Roche Innovation Center Basel, Grenzacherstrasse 124, 4070 Basel, Switzerland. [20]Roche Diagnostics GmbH, Nonnenwald 2, 82377 Penzberg, Germany. [21]ETH Zurich, Department of Chemistry and Applied Biosciences, Vladimir-Prelog-Weg 1-5/10, 8093 Zurich, Switzerland. [22]Cantonal Hospital Baselland, Medical University Clinic, Rheinstrasse 26, 4410 Liestal, Switzerland. [23]University Hospital Basel, Gynecological Cancer Center, Spitalstrasse 21, 4031 Basel, Switzerland. [24]Max Planck ETH Center for Learning Systems, Munich, Germany. [25]University Hospital Basel, Spitalstrasse 21/Petersgraben 4, 4031 Basel, Switzerland. [26]University Hospital Basel, Department of Information- and Communication Technology, Spitalstrasse 26, 4031 Basel, Switzerland. [27]University Hospital Basel, Brustzentrum, Spitalstrasse 21, 4031 Basel, Switzerland. [28]University Hospital Zurich, Department of Medical Oncology and Hematology, Rämistrasse 100, 8091 Zurich, Switzerland. [29]University Hospital Zurich, Department of Gynecology, Frauenklinikstrasse 10, 8091 Zurich, Switzerland. [30]University of Zurich, Services and Support for Science IT, Winterthurerstrasse 190, 8057 Zurich, Switzerland. [31]University of Bern, Department of BioMedical Research, Murtenstrasse 35, 3008 Bern, Switzerland. [32]ETH Zurich, Department of Biology, Wolfgang-Pauli-Strasse 27, 8093 Zurich, Switzerland. [33]Roche Pharmaceutical Research and Early Development, Roche Innovation Center Munich, Roche Diagnostics GmbH, Nonnenwald 2, 82377 Penzberg, Germany. [34]University Hospital Basel, Brustzentrum & Tumorzentrum, Petersgraben 4, 4031 Basel, Switzerland. [35]University Hospital Basel and University of Basel, Department of Surgery, Brustzentrum, Spitalstrasse 21, 4031 Basel, Switzerland. [36]University Hospital and University of Zurich, Department of Neurology, Frauenklinikstrasse 26, 8091 Zurich, Switzerland. [37]University of Zurich, Faculty of Medicine, Zurich, Switzerland. [38]University Hospital Zurich, Rämistrasse 100, 8091 Zurich, Switzerland.

