## [Peer Review File · Nature Communications]

Detection of isoforms and genomic alterations by high-throughput full-length single-cell RNA sequencing in ovarian cancerREVIEWER COMMENTS

Reviewer #1 (Remarks to the Author): expertise in long read computational bioinformatics

The authors use both short-read and long-read scRNA-seq technology to investigate the isoform and genomic features of ovarian cancer. They modify the sequencing protocols and capture plenty of novel isoforms and detect cell type- and cell-specific isoforms at usage and expression level between tumors and control samples. In addition, they also identify novel gene fusions using with the validation from scDNA-seq data. Leveraging the dataset, the author also identifies several key isoforms that may potentially important during EMT. Overall, this study provides a very valuable dataset in the field of ovarian cancer research, and the comparative analysis also highlights the unique advantages of long-length sequencing in single-cell cancer research. It may potentially drive advances in precision medicine and personalized medicine. Below are some concerns and comments.

Major Points:

1. While the authors sequenced many samples and cells by different types of sequencing technologies, they did not make in-depth interpretation of these datasets. Some of the analysis is descriptive and looks superficial. For example, the authors identified COL1A1 tend to use longer 3'UTR in tumor cells than distal cells, but they did not investigate the mechanism by experiment nor make an explanation with published studies. Most isoforms highly expressed in cancer cells tend to have shorter 3'UTRs. I would suggest the authors take a global view of the 3'UTR length alteration between control and tumor cells. Please also make explanations about the candidates the authors chose for further analysis and explore the possible mechanisms.
2. Although there are a lot of interesting points in the dataset, but the authors did not make a comprehensive analysis on those points. For example, in Fig. 2a, we can see that the HGSOC cells seem to have two subpopulations at short-read-based gene-level analysis, but we cannot see that at long-read gene/isoform-level. How do you explain that? The isoform-level analysis have more features than gene-level analysis, maybe we can find some novel subpopulations using isoform expression data. However, it seems the clustering ability of isoform-level is not as good as gene-level. In addition, in Extended Data Fig.4, there are significant heterogeneity in HGSOC cells. Is it possible that different patients have different isoform usage? By exploring the differences (e.g., isoform usage, mutations) between patients, we may get more information for personalized medicine.
3. The overall analysis is not convincing or solid. For example, the detection of novel isoforms is critical for down streaming analysis, the authors should run more parallel analysis to confirm the accuracy of the results. Also, the authors should make a detailed explanation on data processing steps in Methods. For example, what is used for UMAP analysis? Is batch effect removed before clustering? How to integrate different samples together? Furthermore, the authors should also perform more global analysis to compare the differences between tumor and control samples while not just focusing on several key genes.

4. The manuscript needs to be well organized, there needs to be a connection between different parts of the entire article. The paper is not well structured, and some figures may express the same conclusion, such as Fig. 4.

Minor Points:

1. In line 198, it may be better to use the distribution of multi-isoform gene number to compare the isoform diversity.
2. In Fig 1E, what is the sorting basis of bars?
3. In Fig 1F, the total number of isoforms is not 40,046 (Line 147). Does an isoform have two more biotypes?
4. Extended data Fig. 4a-f. Please label the exons that are mentioned in the manuscript.
5. The code in GitHub should include the entire data processing steps of the work.
6. As a data resource, please use a web interface, such as Shiny, to show the data.

Reviewer #2 (Remarks to the Author): expertise in ovarian cancer splicing and isoforms

In this manuscript, Arthur Dondi and co-authors optimize PacBio sequencing strategies and apply their approach to capture cell type-specific genomic and transcriptomic alterations of ovarian cancer patients. This study describes a workflow to increase the sequencing depth of long-read scRNA-seq, which is useful for detection of novel isoforms and genomic alterations. Overall, this is a well-designed study and the manuscript is well written.

The following comments are suggested for consideration by the authors.

1. The authors performed scRNA-seq using five samples including three samples were derived from HGSOC omental metastases and two from matching distal tumor-free omental tissues. Why the authors chose omental tissue other than fallopian tube or ovarian surface epithelium (Both fallopian tube and ovarian surface epithelium are cells-of-origin for high-grade serous ovarian carcinoma, Shuang Zhang, Nat commun. 2019)?
2. The authors claimed that HGSOC cells account for 15% of identified cell types (lines 169-170). Can you explain why HGSOC has a low proportion?
3. The authors identified a novel IGF2BP2::TESPA1 fusion using long-read scRNA and validated by scDNA-seq. Is there any other ovarian cancer patient who have this fusion (DNA sequencing data from other cohorts)?

4. How about the mutations in the single-cell DNA-sequencing data as scDNA-seq is capable of detecting the rarest somatic mutations.

5. Line 226, “resulting in a truncated protein” should be resulting in a truncated transcript or isoform.

Reviewer #3 (Remarks to the Author): expertise in single cell RNA-seq bioinformatics

Dondi et al present a very nice body of work showing the benefits of long-read single-cell sequencing in detecting isoforms and genomic alterations in ovarian cancer. My main concern with the manuscript regards the detection of the novel fusion genes that you have identified and if they are in fact read fusions or artefacts of the library preparation. I lay out my concerns and comments below:

Generally the paper is written in a style that seems to over hype some of the findings. Please tone down the paper conclusions and direction and make your findings more relevant and specific to personalised medicine for ovarian cancer and not cancer as a generalised subject. For example, there is no mention of ovarian cancer in the title and it makes the reader think that you have studied many different types of cancer in this manuscript.

Please provide more QC on the quality of the sequencing. Also how many concatemeric sequences did you achieve per sequencing read.

Extended data Figure 1:

- The UMIs are shifted within the long-read approach compared to the short-read. Why is this? One possible explanation for exploded UMI counts is the suggestion that there are errors within your sequencing in the long-read UMI and this leads to artificial counts.
- Why is there a shift in the genes detected in your data? I assume this is before filtering, what is your QC metrics before and after filtering?
- Aligned with the above two comments, why do you have so many genes detected and why do the numbers not reflect the greater number of genes detected in Extended data Figure 1b?

Line 92-93:

- This is not the case, there are references such as: PMID: 32887687, PMID: 36781734, PMID: 36289342

Line 102:

- Please clarify this, as I believe there are other PacBio datasets that include more than 2,571 cells , e.g. this paper seems to have included 10,000 cells:

<https://www.biorxiv.org/content/10.1101/2021.10.01.462818v>

Figure 5:

- It has been reported that template switching and PCR can generate significant false alternative

transcripts and chimeric artefacts (

<https://www.sciencedirect.com/science/article/pii/S0888754305003770>,

<https://genomebiology.biomedcentral.com/articles/10.1186/gb-2011-12-2-r18>). You do not employ any strategy to determine that the IGF2BP2::TESPA1 is a real fusion event and not just a PCR artefact. I am not convinced that your novel fusion detection approach isn't just detecting chimeric artefacts amplified early during the PCR cycles. No further validation is performed to confirm these fusion events. Your extended figure 5 states this this figure is intended as a validation of the IGF2BP2::TESPA1 breakpoint. However, this is just an IGV view and not an experimental validation.

- In order to be convinced, a strategy for removing chimeric artefacts is required or an experimental validation that does not include PCR.

General response to the reviewers:

First, we would like to thank the reviewers for their valuable inputs, comments, and questions. We addressed all of them below: we have experimentally validated the fusion on the genomic level as requested by Reviewer 3, compared our data to external cohorts and added QC metrics as suggested by Reviewer 2, and performed additional analysis as asked by Reviewer 1. Additionally, thanks to Reviewer 1's suggestion, we have unveiled and described a TGF- β driven microRNA downregulation mechanism that further supports our hypothesis of an epithelial-to-mesenchymal transition of the tumor microenvironment. We feel that altogether the changes greatly benefitted the manuscript and hope that it is now ready for publication.

REVIEWER COMMENTS

Reviewer #1 (Remarks to the Author): expertise in long read computational bioinformatics

The authors use both short-read and long-read scRNA-seq technology to investigate the isoform and genomic features of ovarian cancer. They modify the sequencing protocols and capture plenty of novel isoforms and detect cell type- and cell-specific isoforms at usage and expression level between tumors and control samples. In addition, they also identify novel gene fusions using with the validation from scDNA-seq data. Leveraging the dataset, the author also identifies several key isoforms that may potentially important during EMT. Overall, this study provides a very valuable dataset in the field of ovarian cancer research, and the comparative analysis also highlights the unique advantages of long-length sequencing in single-cell cancer research. It may potentially drive advances in precision medicine and personalized medicine. Below are some concerns and comments.

Major Points:

1. *While the authors sequenced many samples and cells by different types of sequencing technologies, they did not make in-depth interpretation of these datasets. Some of the analysis is descriptive and looks superficial. For example, the authors identified COL1A1 tend to use longer 3'UTR in tumor cells than distal cells, but they did not investigate the mechanism by experiment nor make an explanation with published studies. Most isoforms highly expressed in cancer cells tend to have shorter 3'UTRs. I would suggest the authors take a global view of the 3'UTR length alteration between control and tumor cells. Please also make explanations about the candidates the authors chose for further analysis and explore the possible mechanisms.*

Response: We thank the reviewer for this very helpful comment that encouraged us to perform an additional 3'UTR length analysis, which has led to concrete findings. Reviewer 1 states that “*Most isoforms highly expressed in cancer cells tend to have shorter 3'UTRs.*”, as 3'UTR shortening is one possible mechanism of cancer cells to avoid microRNAs (miRNAs)-mediated mRNA degradation¹. Another known mechanism is the downregulation of miRNAs. If miRNAs are downregulated in cancer cells, their target isoforms with a longer 3'UTR are not degraded anymore, resulting in lengthened 3'UTRs compared to cell types still expressing the miRNAs².

We compared 3'UTR lengths in cancer versus distal cells, and found an overall trend towards 3'UTR lengthening in cancer cells. More context on the global analyses that was performed, as well as the associated figures, are provided in our response to Reviewer 1, Major point 3:

“When testing for differential APA between HGSOV and distal cells, we found shortened 3'UTR in 85 and lengthened 3'UTR in 203 genes (n=4758) (Figure 4f). There was a notable trend toward

lengthening of the 3'UTR in cancer cells ($P=5.59 \times 10^{-20}$), with *COL1A2* emerging once more as the most prominent finding ($P_{\text{corr}}=7.48 \times 10^{-47}$, 61% change) (**Figure 3b-c, Figure 4f**). Expression levels remained consistent between genes featuring either shortened or lengthened 3'UTRs. Furthermore, neither miRNA profiles nor canonical pathways exhibited an overlap exceeding 20% with either the lengthened or shortened gene sets (**Methods**)."

Additionally, 3'UTR differential analysis revealed major differences between mesothelial cells derived from the metastasis (TME) and matched tumor-free (distal) omentum. Briefly, we found multiple genes where isoforms had an lengthened 3'UTR in the TME and demonstrated that this may be due to a TGF- β -driven downregulation of miR-29, leading to an overexpression of genes that promote epithelial-to-mesenchymal transition.

To account for those findings, the entire section "Differential isoform expression in the tumor microenvironment reveals epithelial-to-mesenchymal transition" has been rewritten as shown below. Figure 3 was also changed: previous **Figure 3a,e,f** are now **Extended Data Figure 4a,b,d**, and previous **Figure 3c** is now **Figure 3f**. **Figure 3 a-e** are new.

Tumor microenvironment shows epithelial-to-mesenchymal transition through TGF β -driven miR-29 downregulation

In the subsequent analysis, we compared stromal fibroblasts and mesothelial cells derived from metastasis (TME) and matched tumor-free (distal) omentum. In distal samples, fibroblasts and mesothelial cells formed distinct clusters in both short- and long-read data (**Figure 3a left, Extended Data Figure 5a**). In metastatic samples, however, TME fibroblasts and mesothelial cells formed a bridge in the UMAP embedding, suggesting that these cells might undergo a cell state transition. To test this hypothesis, we conducted gene set enrichment analysis and found that the epithelial-to-mesenchymal transition (EMT) pathway was enriched in TME compared to distal mesothelial and fibroblast cells (**Figure 3a, right**). Similarly, collagen fibril organization and extracellular matrix (ECM) pathways were enriched in TME cells, indicating a reprogramming of the TME cells during metastasis formation (**Extended Data Figure 5b**). Additionally, we compared the alternative poly-adenylation (APA) (**Methods**). Among the 2,876 genes tested for APA, the isoforms of 26 genes in TME mesothelial cells exhibited significant 3'UTR lengthening compared to distal mesothelial cells, while 13 genes showed shortened 3'UTRs (**Figure 3b**). Collagen-encoding genes *COL1A2*, *COL3A1*, *COL5A2*, and *COL6A1* were similarly lengthened and up-regulated in TME cells (**Figure 3c**), with *COL1A2* having the highest effect size ($P_{\text{corr}}=3.42 \times 10^{-67}$, 53% change).

As 3'UTR lengthening is usually associated with a decrease in expression due to microRNA (miRNA) silencing¹, the increased usage of lengthened 3'UTR in TME mesothelial cells suggests that distal cells may express a distinct set of miRNAs not present in TME cells. Collagen-encoding genes are known to be regulated by the miR-29 family in fibroblasts³. Thus, we used miRDB⁴ to predict gene targets for silencing by miR-29a/b/c (**Supplementary Table 2**). Among the 26 isoforms with lengthened 3'UTR, 9 were predicted targets of miR-29a, almost all described as EMT actors (collagens⁵⁻⁸, *KDM6B*⁹, *TNFAIP3*¹⁰, *FKBP5*¹¹ and *RND3*¹²). In contrast, none of the 13 shortened 3'UTR isoforms were predicted to be miR-29 targets (**Figure 3b**). Furthermore, we compared gene expression of TME and distal mesothelial cells (**Supplementary Table 3**), and genes with lengthened 3'UTR isoforms predicted to be silenced by miR-29 were significantly overexpressed ($P = 1.35 \times 10^{-3}$) in TME mesothelial cells compared to the ones not predicted to be silenced by miR-29a (**Figure 3d**).

Mesothelial TME cells also differentially expressed miR-29 targets which are major ECM genes, such as collagen gene *COL1A1* (fold-change=9.0, $P_{\text{corr}}=1.72 \times 10^{-124}$), *MMP2*¹³ (fold-change=4.1, $P_{\text{corr}}=1.51 \times 10^{-30}$), and *LOX*¹⁴ (fold-change=10.6, $P_{\text{corr}}=6.33 \times 10^{-51}$), which is also lengthened ($P_{\text{corr}}=7.07 \times 10^{-2}$, 77% change). Overall, ECM-related genes known to be targeted by miR-29 were upregulated in TME cells compared to distal cells (**Figure 3e**), supporting the hypothesis that the observed EMT was potentially linked with the miR-29 downregulation. When comparing differentially expressed isoforms in TME mesothelial cells to distal cells, *COL1A1* was also the gene with the highest change in relative isoform abundance amongst all its isoforms ($P_{\text{corr}}=6.34 \times 10^{-49}$, 86% usage change, **Methods**) (**Figure 3f**). In TME mesothelial cells, the *COL1A1* canonical 3' poly-adenylation site was used, whereas distal cells used a premature poly-adenylation site, leading to the formation of truncated isoforms (**Figure 3f**). When incorporating only protein-coding isoforms and removing the truncated isoforms from the analysis, the gene expression fold-change increased from 9 to 62-folds ($P_{\text{corr}}=3.15 \times 10^{-183}$). This overexpression of canonical *COL1A1* in the TME can be explained by the absence of miR-29 silencing, as previously described¹⁵.

The miR-29 family is known to be an EMT inhibitor¹⁵. Its silencing through the TGF β pathway correlates with the upregulation of ECM-encoding genes, including multiple collagens, as reported in the present study. The main TGF β gene *TGFB1* was found to be enriched in TME mesothelial cells (fold-change=1.4, $P_{\text{corr}}=2.32 \times 10^{-2}$). Furthermore, in distal mesothelial cells, 38% of *TGFB1* isoforms comprise an alternative 3' exon, leading to aberrant protein expression (**Extended Data Figure 5c**), while the canonical protein-coding *TGFB1* isoform *ENST00000221930.6* is overexpressed in TME cells (fold-change=2.3, $P_{\text{corr}}=6.59 \times 10^{-9}$). miR-29 is also regulated through the expression of non-coding RNAs that act as molecular sponges, directly binding to miR-29 and, therefore, leading to the overexpression of their targets. The TGF β -regulated long non-coding RNA H19, which enhances carboplatin resistance in HGSOC, has been reported to promote EMT through the *H19/miR-29b/COL1A1* axis^{16–18} and was found to be overexpressed in the TME mesothelial cells (fold-change=4.6, $P_{\text{corr}}=3.46 \times 10^{-34}$). Circular RNAs have also been described as miR-29 sponges, notably circMYLK, and circKRT7 in HGSOC. circMYLK and circKRT7 originate from *MYLK* and *KRT7*, respectively, which are both significantly overexpressed in TME mesothelial cells (fold-change=4.1, $P_{\text{corr}}=6.97 \times 10^{-96}$ and fold-change=4.6, $P_{\text{corr}}=2.18 \times 10^{-13}$)^{19,20}. Similarly, TME mesothelial cells expressed the endogenous isoform of *GSN* (cGSN), while distal cells only expressed the secreted isoform (pGSN)²¹ (**Extended Data Figure 5d**). cGSN has been shown to be under TGF- β control in breast cancer and to increase EMT marker expression²². In conclusion, our findings strongly support that, in omental metastases, the mesothelial cells transition into cancer-associated fibroblasts (CAFs), partly through the TGF- β /miR-29/Collagen axis.

Figure 3: Epithelial-to-mesenchymal transition in the tumor microenvironment.

(a) Zoom of UMAP embeddings of the cohorts' long-read - gene level data (Figure 2a, middle column) highlighting tumor and stromal (mesothelial and fibroblast) cells, colored by biopsy tissue type (left) and EMT gene set signal (right). (b) Volcano plot of genes with APA in mesothelial cells. Genes have

either a lengthened (red) or shortened (blue) 3'UTR in TME compared to distal mesothelial cells. Differentially lengthened or shortened genes targeted by miR-29 are colored in green. Genes with $-\log_{10}(\text{p-adjusted}) > 10$ and $|\text{Fraction Change}| > 0.4$ are annotated. **(c)** IGV view of 3'UTR raw coverage of *COL1A2*, *COL6A1*, *COL3A1*, and *COL5A2* in tissue cell types. On the top left between brackets, the coverage range is displayed throughout each condition. In blue, Ensembl canonical 3'UTR, and for each gene, distal (d) and proximal (p) APA sites are annotated. **(d)** Log fold-change expression between TME and distal mesothelial cells of lengthened genes targeted (+, green) or not targeted (-, red) by miR-29, and shortened genes (blue). **(e)** Cohort UMAP embedding long-read data - gene level, colored by gene set signal of ECM-related genes targeted by miR-29. **(f)** ScisorWiz representation of *COL1A1* isoforms. Colored areas are exons, whitespace areas are intronic space, not drawn to scale, and each horizontal line represents a single read colored according to cell types. Dashed boxes highlight the use of the canonical 3' UTR in TME fibroblasts and mesothelial cells, while distal mesothelial cells use an earlier 3' exon termination.

Extended Data Figure 5: Epithelial-to-mesenchymal transition in the tumor microenvironment.

(a) UMAP embeddings of the cohorts' long-read data of short-read data - gene level (left), long-read data - gene level (middle), long-read data - isoform level (right), colored by tissue type. (b) Gene set variation analysis (GSVA) scores for different cell types. Heatmap colors from blue to red represent low to high enrichment. (c) ScisorWiz representation of isoforms in *TGFBI*. Dashed box highlights the non-canonical 3'UTR used by distal mesothelial cells. (d) ScisorWiz representation of isoforms in *GSN*. Dashed boxes highlight the TSS, where mesothelial TME and HGSOc cells differentially express the *cGSN* isoform, while mesothelial distal cells and fibroblasts use *pGSN*.

The “Pathway enrichment analysis” section of the Methods was rewritten as follows:

We used GSVA to perform pathway enrichment analysis. When comparing TME and distal cells, gene sets were obtained from MSigDB²³, except for the miR-29 targets that were obtained from Cushing and colleagues²⁴. When we searched for pathways enriched in lengthened or shortened 3’UTRs in cancer cells, we used the investigate function of the GSEA webpage (<https://www.gsea-msigdb.org/gsea/msigdb/human/annotate.jsp>), and compared against all MSigDB gene sets²³.

The APA analysis section of the Methods was written as follows:

Alternative polyadenylation analysis

To analyze differences between 3’UTR lengths, we used a modified version of DaPars2²⁵ (https://github.com/ArthurDondi/DaPars2_LR), with an APA site detection adapted to long-read coverage.

Briefly, we identified the 3’UTR exon of each isoform and overlapping 3’UTR exons with different 5’ start positions were discarded from analysis, as they create false positive APA. Then, for each remaining 3’UTR we computed the coverage for each cell type. The distal site position was defined as the most 3’ position with a coverage superior to 10 in all cell types:

$$L^* = \max\left(\left\{k: w_k^c > 10, 1 < k < L, 1 < c < m\right\}\right)$$

where L is the length of the annotated 3’ UTR region and L^* is the defined distal site position. w_k^c is the coverage in cell type c at position k , and m is the total number of cell types.

We inferred the exact location of APA site by maximizing the coverage gap between the 50 positions before and after the possible APA sites, based on the two-polyA-site model, the most common model of APA regulation:

$$(C^*, P^*) = \operatorname{argmax}_{1 < c < m, 1 < P < L^*} \left(\frac{1}{50} \sum_{i=1}^{50} w_{P+i}^c - \frac{1}{50} \sum_{i=1}^{50} w_{P-i}^c \right)^2$$

where P is the estimated length of alternative proximal 3’ UTR, and the optimal proximal site P^* in cell type C^* is the one with the maximal objective function value. w_k^c is the coverage at position k and cell type c , and m is the total number of cell types.

Then distal and proximal site coverages W_d and W_p in condition C were defined as:

$$W_d = \frac{1}{50} \sum_{i=1}^{50} w_{L^*-i}^c, \quad W_p = \frac{1}{50} \sum_{i=1}^{50} w_{P^*-i}^c$$

The fraction of distal polyA site usage is then defined as $F = \frac{w_d}{w_d + w_p}$

The degree of difference in APA usage in cell types C_1 and C_2 can be quantified as a *Fraction Change*, which is capable of identifying 3’ UTR lengthening (positive index) or shortening (negative index):

$$\text{Fraction Change} = F_{C_1} - F_{C_2}$$

2. Although there are a lot of interesting points in the dataset, but the authors did not make a comprehensive analysis on those points. For example, in Figure 2a, we can see that the HGSOc cells seem to have two subpopulations at short-read-based gene-level analysis, but we cannot see that at long-read gene/isoform-level. How do you explain that?

Response: The cancer subpopulations seen in short-reads are patient-specific (**Extended Data Figure 2d**). Although the patient-specific cancer clusters are less defined in the UMAPs generated from long-read gene/isoform expression than in the short-reads one, we can still see in **Extended Data Figure 2d** the separation by patients inside the cluster.

There was a mislabelling of the patients in **Extended Data Figure 2d**. It is now corrected as shown below, and we added the cluster names. We apologize and we hope the correction will help answer the reviewer's question.

Extended Data Figure 2: Cell type marker detection and embedding by sample. [...] (d) UMAP embeddings of the cohort, colored by sample ID for short-read - gene level (left), long-read - gene level (middle), and long-read - isoform level (right) data.

The isoform-level analysis have more features than gene-level analysis, maybe we can find some novel subpopulations using isoform expression data. However, it seems the clustering ability of isoform-level is not as good as gene-level.

Response: The isoform expression matrix contains more entries, as we observe more isoforms than genes, and more values are equal to 0 as we have limited coverage due to sequencing costs. While we indeed believe that the isoform data has the potential to reveal novel or different subpopulations, here the limited coverage results in a bridge of low read count cells between cancer and mesothelial clusters. This is evident from the isoform counts per cell, as illustrated in the figure below.

Rebuttal Figure 1: UMAP cohort visualization based on isoforms expression, colored by number of isoforms expressed per cell.

In addition, in Extended Data Figure 4, there are significant heterogeneity in HGSOc cells. Is it possible that different patients have different isoform usage? By exploring the differences (e.g., isoform usage, mutations) between patients, we may get more information for personalized medicine.

Response: Patients with High-Grade Serous Ovarian Cancer (HGSOc) typically share *TP53* mutations as a common genetic characteristic. In addition, the mutations (SNVs) we have identified were previously recorded in the panel DNA sequencing data for all three patients and were patient-specific. Please refer to **Supplementary Table 1** for comprehensive details regarding these mutations. We also investigated patient-specific differences in gene fusions and found the patient-specific *IGF2BP2::TESPA1* fusion. Furthermore, the isoform diversity observed in previous **Extended Data Figure 4** is often shared between patients, and observed at such low frequency that it is hard to draw any conclusions.

We performed differential isoform usage analysis and found that differences between patients were minimal. However, we found one important difference between Patient 3 and the two other patients in the *OAS1* gene (more details can be found below in Major Point 3, Reviewer 1, section “Differential isoform expression in cancer reveals isoform-specific usage of EMT genes”):

Although isoforms differentially expressed in cancer cells were similar among patients, there was one significant case of patient-specific expression. For *OAS1*, patient 3 predominantly expressed isoform *p42*, while patients 1 and 2 exhibited a balanced distribution of isoforms *p42* and *p46*. The *p42* and *p46* expressions are known to be allele-specific, caused by the rs10774671 SNP, a splice acceptor A/G variation. However, this cannot explain the different expression levels as all patients have both the A and G alleles (**Figure 4e**). Given that isoform *p42* is more susceptible to Nonsense-Mediated mRNA Decay (NMD) and possesses diminished enzymatic activity²⁶, the observed differences could potentially indicate a diminished *OAS1* activity in Patient 3. Whether *OAS1* has an impact on ovarian cancer is still to be elucidated.

The rare differences between patients could be due to the disparity of gene expression. For example, when we compared isoform usage in cancer and distal cells, we found that, in *CERK* and *RTN1*, cancer-specific isoform usage was also patient specific. Those results do not appear when comparing isoform usage between patients, because the other patients had no *CERK* or *RTN1* expression at all,

and the genes were discarded from the analysis (more details can be found below in Major Point 3, Reviewer 1, section “Differential isoform expression in cancer reveals isoform-specific usage of EMT genes”):

In *RTN1*, distal cells expressed the isoform RTN1-A and RTN1-B, while Patient 3’s cancer cells expressed RTN1-C, an isoform known to bind to the anti-apoptotic protein Bcl-xL and reduce its activity [...]. For *CERK*, by contrast, patient 2’s HGSOC cells strongly expressed a novel isoform leading to a shortened protein (**Extended Data Figure 6g**).

3. The overall analysis is not convincing or solid. For example, the detection of novel isoforms is critical for down streaming analysis, the authors should run more parallel analysis to confirm the accuracy of the results.

Response: We used SQANTI for novel isoform detection, classification, and artefact filtering, as this is the gold standard within the field (<https://github.com/ConesaLab/SQANTI3>). In addition to the SQANTI output, we validated the 5’ and 3’ completeness of the novel isoforms by comparing them to experimental CAGE/PolyA peak databases (FANTOM5 and PolyASite). We also sent the list of novel detected isoforms for external validation to the researchers maintaining the GENCODE database, which is the most intensely curated database for isoforms. GENCODE validated and characterized the biotype of ~80% of the novel isoforms we detected. Throughout, we have taken rigorous steps to ensure the accuracy of our results with gold-standard procedures (SQANTI), validation with experimental CAGE/PolyA databases and an independent group of experts (GENCODE). Thus, we are convinced that the reported results are solid.

Also, the authors should make a detailed explanation on data processing steps in Methods. For example, what is used for UMAP analysis? Is batch effect removed before clustering? How to integrate different samples together?

Response: We apologize for this omission, and we now provide all details. In brief, gene/isoform counts were normalized with *sctransform*, regressing out cell cycle effects, library size, and sample effects as non-regularized dependent variables. Further details are below.

For short reads, we modified corresponding subsection in **Material and Methods** as follows:

10x Genomics read data processing, normalization and visualization.

Expression data of each sample was analyzed using the *scAmp* workflow²⁷. In brief, UMI counts were quality-controlled and cells and genes were filtered to remove mitochondrial and ribosomal contaminants. Cells for which over 50% of the reads mapped to mitochondrial genes and cells with fewer than 400 genes expressed were removed. **By default, all non-protein-coding genes, genes coding for ribosomal and mitochondrial proteins, and genes that were expressed in less than 20 cells were removed.** Doublet detection was performed using *scDbIFinder*²⁸. **Subsequently, counts were normalized with *sctransform*²⁹, regressing out cell cycle effects, library size, and sample effects as non-regularized dependent variables.** Similar cells were grouped based on unsupervised clustering using *Phenograph*³⁰, and automated cell type classification was performed independently for each cell³¹ using gene lists defining highly expressed genes in different cell types. Major cell type marker lists were developed in-house based on unpublished datasets (manuscripts in preparation), including the Tumor Profiler Study³², using the *Seurat FindMarkers* method³³. Immune subtype marker gene lists were obtained from Newman *et al.*³⁴ and enriched with T cell subtypes from Sade-Feldman *et*

*al.*³⁵ The results of the unsupervised clustering and cell typing are visualized in a low-dimensional representation using Uniform Manifold Approximation and Projection (UMAP).

For long reads, we also added this paragraph:

Normalization and visualization

Long read gene expression counts were normalized and visualized as described above for short reads. Long read gene expression counts were normalized using 10,000 features instead of the default 3,000 in `sctransform`²⁹.

Furthermore, the authors should also perform more global analysis to compare the differences between tumor and control samples while not just focusing on several key genes.

Response: We compared tumor and control samples on two levels: tumor cells with all distal cells, and metastasis TME with corresponding distal cell types (with a focus on mesothelial cells). The additional analyses and findings regarding the metastasis TME mesothelial cells are reported in our response to Reviewer 1's Major Point 1.

For the comparison of cancer with distal cells, we first analyzed the distributions of cell type-specific novel isoforms, revealing that cancer cells expressed more novel isoforms (**Extended Data Figure 4**). We also performed differential isoform usage analysis between cancer and distal cells and focused on the most differentially expressed genes (**Figure 4**). Focusing on these genes was necessary as isoforms are understudied and insufficiently annotated (even protein-coding ones), such that extensive bibliographical research was needed to assign biological meaning to specific isoform expressions. We were able to do this for *IGF1 Class III*, *cGSN*, *TPM2.3*, *RTN1-C*, etc., but not for *CERK* or *ICAM3* due to current lack of functional isoform characterization.

In this revision, we also performed novel global analyses between cancer and distal cells. Thanks to the help of GENCODE, isoform biotypes were available even for novel isoforms in our datasets, and we were able to compare biotypes of isoforms differentially expressed between cancer and distal cells. Isoform-level analysis accounting for non-coding isoforms also revealed that protein-coding gene expression was overestimated by 20% on average. We also performed alternative poly-adenylation (APA) analysis between cancer and distal cells, as stated in our response to Reviewer 1's Major Point 1.

Figure 4 and **Extended Data Figure 4** (now **Extended Data Figure 6**) were also redone to account for the novel analyses performed and to remove any redundancy, in line with Reviewer 1's demand in Major Point 4.

You can find the results of our comparison of cancer against distal cells below:

Differential isoform usage in cancer reveals changes in biotypes

After comparing cells from the TME with distal cells, we investigated which isoforms, biotypes, and poly-adenylation sites were differentially used between cancer and all distal cells. HGSOC cells expressed isoforms differentially with a change in relative isoform abundance of more than 20% in 960 genes (15.1%), compared to all distal cells (6,353 genes tested in total, **Fig. 4a, Supplementary Table 4, Methods**). In 36% of those 960 genes, the highest expressed isoform biotype changed between conditions (**Fig. 4b**). In 32% of instances, there was a transition from a protein-coding to a non-protein-coding isoform, and in 17% of cases, cancer cells expressed a protein-coding isoform

while distal cells expressed a non-protein-coding isoform. Only 39 genes (0.6%) had an isoform switch with a change in relative isoform abundance of more than 50%. 59% of these switched isoforms demonstrated a biotype transition (49% of protein-coding to non-protein coding transition, **Fig. 4c**), and in 33% of cases, cancer cells expressed a protein-coding isoform while distal cells expressed a non-protein-coding isoform. Additionally, in cancer, distal and TME cells, on average 20-21% of the expression in protein-coding genes was non-coding (**Extended Data Fig. 6a**), and 13-14% of protein-coding genes had more than 50% of non-coding expression. This means that, on average, using only gene-level information to estimate protein expression (as done in short-read data) will lead to an overestimation of 20%.

The ten genes with the statistically most significant switches were *IGF1*, *TPM2*, *NCALD*, *VAMP8-VAMP5*, *EXOSC7*, *ICAM3*, *CERK*, *OBSL1*, *GSN*, and *RTN1* (**Supplementary Table 4, Methods**). In *IGF1*, cancer cells across all patients predominantly used the second exon of the gene as their transcription start site (secreted isoform, *Class II*), whereas non-cancerous cells primarily used the first exon (endogenous isoform, *Class I*)³⁶ (**Fig. 4d**). On the contrary, similarly to TME mesothelial cells, cancer cells expressed the endogenous isoform of *GSN* (c*GSN*), while distal cells only expressed the secreted isoform (p*GSN*)²¹ (**Extended Data Fig. 5d**). In *RTN1*, distal cells expressed the isoform *RTN1-A* and *RTN1-B*, while Patient 3's cancer cells expressed *RTN1-C*, an isoform known to bind to the anti-apoptotic protein Bcl-xL and reduce its activity^{37,38} (**Extended Data Fig. 6b**). In the tropomyosin gene *TPM2*, which is involved in TGF- β -induced EMT, cancer cells differentially expressed exon 6b (isoform *TPM2.3*, expressed in epithelial cells³⁹) and the alternative 3'UTR exon 9a (**Extended Data Fig. 6c**). In *VAMP5*, the overexpressed isoform in HGSOC cells was a *VAMP8-VAMP5* read-through gene, i.e., a novel gene formed of two adjacent genes, previously described in human prostate adenocarcinoma⁴⁰ (**Extended Data Fig. 6d**). HGSOC cells expressed almost no wild-type (wt) *VAMP5* but had a significantly higher *VAMP8* expression than distal cells ($P_{\text{corr}}=1.0 \times 10^{-15}$), indicating that this read-through gene was under transcriptional control of *VAMP8*. With a short-read 3' capture method, this *VAMP8-VAMP5* expression cannot be distinguished from the wt *VAMP5* expression. For *NCALD* and *OBSL1*, only cancer cells expressed canonical protein-coding isoforms, while other cells expressed short non-coding isoforms (**Extended Data Fig. 6e,f**). For *CERK*, by contrast, patient 2's HGSOC cells strongly expressed a novel isoform leading to a shortened protein (**Extended Data Fig. 6g**). Finally, in *ICAM3*, cancer cells mainly expressed a short protein-coding isoform, while distal cells (mainly T cells) expressed the canonical isoform. More characterization of those isoforms will be necessary in the future to explore the biological implications linked to their expression (**Extended Data Fig. 6h**).

Although isoforms differentially expressed in cancer cells were similar among patients, there was one significant case of patient-specific expression. For *OAS1*, patient 3 predominantly expressed isoform *p42*, while patients 1 and 2 exhibited a balanced distribution of isoforms *p42* and *p46*. The *p42* and *p46* expressions are known to be allele-specific, caused by the rs10774671 SNP, a splice acceptor A/G variation. However, this cannot explain the different expression levels as all patients have both the A and G alleles (**Fig. 4e**). Given that isoform *p42* is more susceptible to Nonsense-Mediated mRNA Decay (NMD) and possesses diminished enzymatic activity²⁶, the observed differences could potentially indicate a diminished *OAS1* activity in Patient 3. Whether *OAS1* has an impact on ovarian cancer is still to be elucidated.

When testing for differential APA between HGSOc and distal cells, we found shortened 3'UTR in 85 and lengthened 3'UTR in 203 genes (n=4758) (Fig. 4f). There was a notable trend toward lengthening of the 3'UTR in cancer cells ($P=5.59 \times 10^{-20}$), with *COL1A2* emerging once more as the most prominent finding ($P_{\text{corr}}=7.48 \times 10^{-47}$, 61% change) (Fig. 3b-c, Fig. 4f). Expression levels remained consistent between genes featuring either shortened or lengthened 3'UTRs. Furthermore, neither miRNA profiles nor canonical pathways exhibited an overlap exceeding 20% with either the lengthened or shortened gene sets (Methods).

Figure 4: Differential isoforms and 3'UTR lengths in cancer.

(a) Number of genes with change in isoform usage between HGSOc and all distal cells. In orange, genes with differentially expressed isoforms and a change in relative isoform abundance >20% (>50% in green). In blue, genes with no differentially expressed isoforms or change in relative isoform abundance <20%. (b) Alluvial plot of biotypes of most expressed isoforms in HGSOc and distal cells in genes containing an isoform change >20% (n=960). Each vein represents the conversion of one biotype to another. For example, in 7 genes, the most expressed isoform in HGSOc cells is protein-coding, while distal cells' one is non-protein-coding. (c) Alluvial plot of biotypes of most expressed isoforms in HGSOc and distal cells in genes containing an isoform switch (>50% change, n=39). (d) ScisorWiz representation of isoforms in *IGF1*, each horizontal line represents a single isoform colored according to cell types. Exons are numbered according to the Gencode reference, Class I and II isoforms are isoforms with starting exons 1 and 2, respectively. (e) Top: IGV view of *OAS1* expression in patients. Patient 3 has low p46 expression compared to others. Bottom: zoom on the last exon of isoform p46, where all patients have at least one mutated A allele in the splice

acceptor site. **(f)** Volcano plot of genes with APA in cancer versus distal cells. Genes have either a lengthened (red) or shortened (blue) 3'UTR in cancer cells compared to all distal cells. Differentially lengthened or shortened genes targeted by miR-29 are colored in green. Genes with $-\log_{10}(\text{p-adjusted}) > 10$ and $|\text{Fraction Change}| > 0.5$ are annotated.

Extended Data Figure 6. Top differentially expressed genes between HGSOc and distal cells. (a) Boxplot of the fraction of non-coding expression in protein coding genes, where each point is a protein-coding gene. ScisorWiz representation of isoforms in (b) *RTN1*, (c) *TPM2*, (d) *VAMP5-VAMP8*, (e) *NCALD*, (f) *OBSL1*, (g) *CERK*, (h) *ICAM3* in cancer, distal and TME cells. Each horizontal line represents a single isoform colored according to cell types. Notable reference isoforms from GENCODE are named on the bottom of each gene.

Relative Methods were added as follows:

Biotype switching analysis

In genes with differential isoform usage between conditions, we compared the biotypes of the most expressed isoform of each condition, and if they were not identical this was considered a change in biotype.

Protein-coding gene expression

Considering only genes with protein-coding biotype (GENCODE) with a minimal expression of 20 UMIs in a condition, we assessed the ratio of non-coding isoforms (NMD, intron retention, etc.) expression to total gene expression (non-coding and coding).

4. *The manuscript needs to be well organized, there needs to be a connection between different parts of the entire article. The paper is not well structured, and some figures may express the same conclusion, such as Figure 4.*

Response: We have further improved the writing and organization of the manuscript, which was assessed very favorably by Reviewer 2. We agree with the comment on **Figure 4**, which we restructured and rewrote the corresponding text.

Figure 4 and **Extended Data Figure 4** (now **Extended Data Figure 6**) were redone to remove any redundancy and add more analysis, in the line of Reviewer 1's demand for more global analysis, and the corresponding section was rewritten (see above). We also rewrote parts of the discussion, as follows:

Discussion

[...]

Taking advantage of the long-read technology, our study provides evidence that cancer cells may induce EMT in TME mesothelial cells. In omental metastases, it has been shown that HGSOC-secreted TGF- β triggers EMT and converts TME cells, including mesothelial cells, into CAFs, which in turn may favor tumor cell adhesion, invasion, and proliferation⁴¹⁻⁴³. Our findings provide further evidence of this phenomenon and reveal that the process might partly be controlled through the TGF- β /miR-29/Collagen axis: secreted and endogenous TGF- β downregulates miR-29 expression, thus increasing the expression of its targets, including collagens⁴⁴. Coincidentally, Han and colleagues¹³ recently showed that omental CAF-derived exosomes from HGSOC patients contained significantly lower miR-29c than normal fibroblasts. Furthermore, the reduced levels of miR-29 have been demonstrated to play a role in the development of cisplatin resistance by upregulating collagen expression⁴⁵.

In addition to TGF- β , HGSOC cells also promote EMT and CAF activation through the secretion of other growth factors such as insulin-like growth factors (IGFs)⁴⁶. In our data, HGSOC cells revealed a profoundly modified IGF system among all patients with a drastic switch from endogenous *Class I* to secreted *Class II IGF1* isoform, *IGF2* overexpression, and a highly expressed *IGF2BP2::TESPA1* gene fusion in one patient. Secreted IGF1 (Class II) and IGF2 activate EMT through IGF1R, by triggering an uncontrolled wound healing response⁴⁷. The *IGF* gene family also promotes cancer growth, survival, proliferation, and drug resistance through signaling via *PI3K-AKT* or *MAPK*, and is a known clinical target in ovarian cancer⁴⁸.

Additionally, we demonstrated the potential of the technology in terms of coverage and sequencing accuracy to detect mutations and gene fusions. In particular, in one patient, the novel fusion *IGF2BP2::TESPA1* was highly overexpressed compared to wt *IGF2BP2* (~10x more) and *TESPA1* (~150x more). *IGF2BP2* is known to be regulated via 3'UTR miRNA silencing⁴⁹, however, the *IGF2BP2::TESPA1* fusion has the unregulated 3'UTR of *TESPA1*, which could explain its overexpression. *TESPA1* is normally expressed in T cells⁵⁰ and long-read data confirmed T cell-specific wt *TESPA1* expression. Short read data however erroneously reported *TESPA1* as the most differentially expressed gene in cancer cells, resulting from 3' end capture of the fusion transcripts. This highlights that short-read scRNA-seq data fails to distinguish between gene and fusion expression, potentially leading to wrong biological conclusions.

In addition to their canonical mRNA isoforms, numerous protein-coding genes express non-coding RNA isoforms, which are wrongly accounted for as protein-coding on the gene expression level and can serve as miRNA sponges and competing endogenous RNAs, especially in EMT and metastasis^{51,52}. In our data, we found that 20% of the protein-coding gene expression was non-coding and in the genes exhibiting a significant isoform switch between cancer and healthy cells, nearly half (49%) transitioned from a protein-coding to a non-coding isoform. Furthermore, 51% of the UMIs were composed of intra-priming and non-canonical artifacts before filtering, and their detection was only possible through isoform classification. Overall, our findings highlight the need for an isoform-specific quantification to accurately assess protein-coding gene expression⁵³ and narrow the RNA-protein gap⁵⁴⁻⁵⁷. Additionally, a better isoform characterization is needed to understand their biological implications, as we partly demonstrated with *IGF1 Class I/II*, *cGSN*, *TPM2.3*, *RTN1-C*, and *OAS1 p42*. [...]

Minor Points:

1. In line 198, it may be better to use the distribution of multi-isoform gene number to compare the isoform diversity.

Response: While isoforms per genes per cell type is an interesting metric, it is not properly normalized: we are more likely to capture more isoforms in cell types that express more reads, and this metric does not account for that. This is why we came up with the reads per isoform metric: it shows that cancer cells express less reads per isoform than mesothelial cells, for example, and that the abundance of isoforms in cancer cells is not only due to overall expression. We nevertheless tried to investigate the mean number of isoforms expressed per gene in each cell.

Please find below the number of isoforms per gene per cell type as **Extended Data Figure 3b**. Previous **Extended Data Figure 3b** became **3a**, and previous **3a** (genes per cells) was removed as it offers low information.

Extended Data Figure 3: QC metrics by cell type in long-read sequencing. (a) Number of isoforms detected per cell, by cell type. (b) Number of isoforms detected per gene, by cell type. (c) Number of unique reads per cell in long-read s, by cell type. (d) Ratio of long reads per isoform in each cell, by cell type. [...]

And we changed the text accordingly:

We analyzed the expression of cell type-specific isoforms. HGSOc cells expressed more isoforms (overall and per gene), and RNA molecules than other cell types (Extended Data Figure 4a-c).

2. In Fig 1E, what is the sorting basis of bars?

Response: The original idea was to sort by level of interest, with first our data and then GENCODE. Thank you for inquiring about the sorting, we can see now that the ordering can be confusing.

Figure 1e is now sorted in descending order, as you can see below.

Figure 1: Study design and long read data overview. [...] **(e)** Percentage of isoforms for which transcription start site is supported by CAGE (FANTOM5) data and transcription termination site is supported by polyA (PolyASite) data, per isoform structural categories. GENCODE.all indicates all protein-coding isoforms in the GENCODE database, GENCODE.FL is a subset of GENCODE.full containing only isoforms tagged as full-length, and GENCODE.MANE is a hand-curated subset of canonical isoforms, one per human protein coding locus. [...]

3. In Fig 1F, the total number of isoforms is not 40,046 (Line 147). Does an isoform have two more biotypes?

Response: 43,741 of the novel isoforms we identified were validated by GENCODE, with 40,046 being considered as novel by their team and 3,695 extended versions of existing isoforms.

We changed the corresponding section in the manuscript for a better understanding of **Figure 1f** :

A total of 52,884 novel isoforms were complete (NIC+NNC), of which 40,046 were confirmed as valid novel isoforms by GENCODE (corresponding to 17% of the current GENCODE v36 database), and 3,695 were extended versions of existing isoforms.

4. Extended data Figure 4a-f. Please label the exons that are mentioned in the manuscript.

Response: We thank the reviewer for pointing out this omission, which is now fixed. You can find the changes in response to Reviewer 1, Major Point 3, **Extended Data Figure 6**.

5. The code in GitHub should include the entire data processing steps of the work.

Response: We added post-processing code to GitHub. We also created a separate GitHub for 3'UTR APA analysis: https://github.com/ArthurDondi/DaPars2_LR

6. As a data resource, please use a web interface, such as Shiny, to show the data.

Response: The sequencing data is confidential patient data, and thus cannot be made available on a public web interface. It is available on demand at EGA (EGAC00001003014).

Reviewer #2 (Remarks to the Author): expertise in ovarian cancer splicing and isoforms

In this manuscript, Arthur Dondi and co-authors optimize PacBio sequencing strategies and apply their approach to capture cell type-specific genomic and transcriptomic alterations of ovarian cancer patients. This study describes a workflow to increase the sequencing depth of long-read scRNA-seq, which is useful for detection of novel isoforms and genomic alterations. Overall, this is a well-designed study and the manuscript is well written.

The following comments are suggested for consideration by the authors.

1. *The authors performed scRNA-seq using five samples including three samples were derived from HGSOC omental metastases and two from matching distal tumor-free omental tissues. Why the authors chose omental tissue other than fallopian tube or ovarian surface epithelium (Both fallopian tube and ovarian surface epithelium are cells-of-origin for high-grade serous ovarian carcinoma, Shuang Zhang, Nat commun. 2019)?*

Response: The aim of the study was not only to use long-read sequencing to study tumor-intrinsic features, such as isoform use and genomic alterations, but also to investigate microenvironment alterations. Since the omentum is the preferred site of HGSOC metastasis, we chose omental metastases biopsies to study changes in the microenvironment and sampled metastatic and distal tumor-free sites allowing for differential comparison of cell type gene and isoform composition.

2. *The authors claimed that HGSOC cells account for 15% of identified cell types (lines 169-170). Can you explain why HGSOC has a low proportion?*

Response: This fraction is the average over all cells from all samples (including 2 healthy omentum samples out of the 5 total samples). As shown in **Extended Data Figure 3b** (previously 2a), Patient 2 Tumor sample has ~50% HGSOC cells, Patient 1 Tumor sample has ~20% HGSOC cells, and Patient 3 Tumor sample has <10% HGSOC cells.

3. *The authors identified a novel IGF2BP2::TESPA1 fusion using long-read scRNA and validated by scDNA-seq. Is there any other ovarian cancer patient who have this fusion (DNA sequencing data from other cohorts)?*

Response: To investigate the presence of the *IGF2BP2::TESPA1* fusion in other ovarian cancer patients, we analyzed the Ovarian Cancer TCGA data expression of *IGF2BP2* and *TESPA1*, on the gene and exon level. We analyzed the exon expression at the breakpoint between exons 4 and 5 for *IGF2BP2*, and between the last coding exon and 3'UTR for *TESPA1* (**Extended Data Figure 9a,b**). If a patient expressed *IGF2BP2::TESPA1* as strongly as it is expressed in patient 2 in our data, the expression between the exons should not correlate. For example, *IGF2BP2* exon 4 is expressed by both the fusion and wild-type genes, while exon 5 is only expressed in the wild-type gene. In Patient 2, exon 4 is >10x stronger expressed than exon 5, because the fusion is 10x more expressed than wild-type *IGF2BP2*. This does not seem to be the case in TCGA data (**Extended Data Figure 9a**). All patients (points) have a similar exon 4 and 5 expression in *IGF2BP2* ($R>0.99$). We observe similar results with *TESPA1* ($R>0.88$) with a weaker correlation because *TESPA1* is sparsely expressed in ovarian cancer samples (**Extended Data Figure 9b**). Moreover, there was no clear correlation between the gene expression of *IGF2BP2* and *TESPA1* in patients from the TCGA database, and no patient had a clear high-*TESPA1* high-*IGF2BP2* expression indicating a potential *IGF2BP2::TESPA1* presence (*TESPA1* expression is low in all patients, max TPM = 4.834, **Extended Data Figure 9c**). It is however important to note that those are bulk measures, and a fusion expression could be hidden by low tumor content or only present in a rare subclone. In conclusion, we found no evidence of *IGF2BP2::TESPA1* presence in bulk RNA expression data in TCGA ovarian cancer patients.

The comparison with the TCGA has been incorporated as **Extended Data Figure 9**, as well as the corresponding text below:

In the ovarian cancer TCGA RNA dataset, the expression levels of exons surrounding breakpoints of *IGF2BP2* and *TESPA1* did not change (**Extended Data Fig. 9a,b**), and the overall expression of the genes did not correlate in any patient, suggesting that we detected an uncommon, patient-specific fusion (**Extended Data Fig. 9c**).

Extended Data Figure 9: Detection of the *IGF2BP2::TESPA1* fusion in the ovarian cancer TCGA data.

(a) TCGA expression of *IGF2BP2* exons 4 and 5, surrounding the *IGF2BP2::TESPA1* breakpoint. **(b)** TCGA expression of *TESPA1* last coding exon and 3'UTR exon, surrounding the *IGF2BP2::TESPA1* breakpoint. **(c)** Log2 normalized gene expressions of *IGF2BP2* and *TESPA1* in TCGA.

4. How about the mutations in the single-cell DNA-sequencing data as scDNA-seq is capable of detecting the rarest somatic mutations.

Response: scDNA-seq is capable of detecting rare somatic SNVs in regions with high coverage, for example, when using multiple displacement amplification and panel or whole-exome sequencing. Here, we used whole-genome sequencing (WGS) at very shallow coverage, as the fusion breakpoint was in an intron (10X Chromium Single Cell CNV Solution, discontinued). WGS at such low coverage (<0.05x) is not suited for SNV detection.

5. Line 226, "resulting in a truncated protein" should be resulting in a truncated transcript or isoform.

Response: We thank the reviewers for pointing this out. While it is indeed the isoform which is truncated, the coding sequence is truncated, too, and if translated, the resulting protein would also be truncated. This has been clarified throughout the manuscript.

Reviewer #3 (Remarks to the Author): expertise in single cell RNA-seq bioinformatics

Dondi et al present a very nice body of work showing the benefits of long-read single-cell sequencing in detecting isoforms and genomic alterations in ovarian cancer. My main concern with the manuscript regards the detection of the novel fusion genes that you have identified and if they are in fact read fusions or artefacts of the library preparation. I lay out my concerns and comments below:

Generally the paper is written in a style that seems to over hype some of the findings. Please tone down the paper conclusions and direction and make your findings more relevant and specific to personalised medicine for ovarian cancer and not cancer as a generalised subject. For example, there is no mention of ovarian cancer in the title and it makes the reader think that you have studied many different types of cancer in this manuscript.

Response: We thank the reviewer for their overall positive feedback on our manuscript. As requested, we tried to make our findings more relevant and specific to ovarian cancer, by changing the title and abstract, and focusing parts of the manuscript (see complete response to Reviewer 1, Major Point 1, above), as well as the discussion on EMT in ovarian cancer.

We changed the title to “Detection of isoforms and genomic alterations by high-throughput full-length single-cell RNA sequencing in ovarian cancer”.

The abstract was also changed as follows :

Abstract

Understanding the complex background of cancer requires genotype-phenotype information in single-cell resolution. Here, we perform long-read single-cell RNA sequencing (scRNA-seq) on clinical samples from three ovarian cancer patients presenting with omental metastasis and increase the PacBio sequencing depth to 12,000 reads per cell. Our approach captures 152,000 isoforms, of which over 52,000 are novel. Isoform-level analysis accounting for non-coding isoforms reveals 20% overestimation of protein-coding gene expression on average. We also detect cell type-specific isoform and poly-adenylation site usage in tumor and mesothelial cells, and find that mesothelial cells transition into cancer-associated fibroblasts in the metastasis, partly through the TGF- β /miR-29/Collagen axis. Furthermore, we identify gene fusions, including an experimentally validated *IGF2BP2::TESPA1* fusion, which is misclassified as high *TESPA1* expression in matched short-read data, and call mutations confirmed by targeted NGS cancer gene panel results. With these findings, we envision long-read scRNA-seq to become increasingly relevant in oncology and personalized medicine.

We also changed the last paragraph of the introduction to reflect on our new findings and focus more on ovarian cancer:

Introduction

[...]

Here, for the first time, we used high-quality, high-throughput long-read scRNA-seq to capture cell type-specific genomic and transcriptomic alterations in clinical cancer patients. We applied both short-read and long-read scRNA-seq to five samples from three HGSOC patients, comprising 2,571 cells, and generated the PacBio scRNA-seq dataset with the deepest coverage to date. We were able to identify over 150,000 isoforms, of which a third were novel, as well as novel cell type-specific

isoforms. Isoform-level analysis revealed that, on average, 20% of the protein-coding gene expression was non-coding, leading to an overestimation of the protein expression. By combining differential isoform and poly-adenylation site usage analysis between cells from the metastatic TME and distal omental biopsies, we found evidence that in omental metastases, mesothelial cells transition into CAFs, partly through the TGF- β /miR-29/Collagen axis. Additionally, we discovered dysregulations in the insulin-like growth factor (IGF) network in tumor cells on the genomic and transcriptomic levels. Thereby, we demonstrated that scRNA-seq can capture genomic alterations accurately, including cancer- and patient-specific germline and somatic mutations in genes such as *TP53*, as well as gene fusions, including a novel *IGF2BP2::TESPA1* fusion.

Please provide more QC on the quality of the sequencing. Also how many concatemeric sequences did you achieve per sequencing read.

Response: We appreciate the concerns on sequencing QC. The number of reads passing each (pre)processing step are provided in **Extended Data Table 1**, and we added a new figure (**Extended Data Figure 1**) providing the concatemeric sequences statistics as well as reads lengths and the filtering of reads attached to based on :

This allowed the generation of a total of 212 Mio HiFi reads in 2,571 cells, which, after demultiplexing, deduplication, and **artifact** removal, resulted in 30.7 Mio unique molecular identifiers (UMIs), for an average of 12k UMIs per cell. (**Extended Data Table 1**). **There was a mean of four cDNA molecules concatenated per sequencing read overall, and cDNA length was similar across samples (Extended Data Fig. 1a,b)**. Artifact removal revealed that filtered 51% of the UMIs were attached to , and it included the removal of intra-priming (63%), non-canonical isoforms (36%), and reverse transcriptase switching (1%)⁵⁸ (**Extended Data Fig. 1c**). It must be emphasized that those artifact reads emerge from the single-cell library preparation and are also present in short reads, where they cannot be filtered and are hence accounted for as valid reads.

Extended Data Figure 1: QC metrics long reads. (a) Histogram showing the number of concatemeric cDNA molecules per sequencing read by sample. (b) Length distribution of long reads after unconcatenation, per sample. (c) Number of UMIs belonging to isoforms passing all filters (full-length non-chimeric, FLNC) and UMIs belonging to filtered isoforms (intra-priming, non-canonical isoform, reverse transcriptase (RT) switching), following SQANTI classification.

Extended data Figure 1:

- The UMIs are shifted within the long-read approach compared to the short-read. Why is this? One possible explanation for exploded UMI counts is the suggestion that there are errors within your sequencing in the long-read UMI and this leads to artificial counts.

Response: There was a labeling error in **Extended Data Figure 1c** (now **Extended Data Figure 2c**) that might have misled the reviewer's interpretation on the blue color corresponding to long reads. This has now been corrected (see **Extended Data Figure 2c** below), and we apologize for the confusion. There are less long-read than short-read UMIs per cell, because long-read sequencing yields fewer reads. In this configuration, the per-base quality of long reads surpasses that of short reads overall, and UMIs errors are rare. Additionally, barcode correction was executed using the IsoSeq3 deduplication tool. Please find below the corrected text and figure:

The short- and long-read datasets were of similar sequencing depth with a median of 4,930 and 2,750 UMIs per cell, respectively (average 10,235 and 6,413 UMIs, **Extended Data Fig. 2a**). Long-read data contained slightly fewer detected genes, and genes detected in both datasets overlapped by 86.4% (**Extended Data Fig. 2b,c**). Paradoxically, the genes detected were overall shorter in long-reads than short-reads, likely due to the concatenation step (**Extended Data Fig. 2d**).

Extended Data Figure 2: QC metrics comparison of short and long reads. (a) Distribution of single UMI reads and (b) genes detected per cell in short- (light-blue) and long-read (light-red) technologies. (c) Overlap of total detected genes between short- and long-read datasets. (d) Gene length distribution of genes detected in short- and long-read technologies, where gene length equals the sum of exon length of the longest isoform associated with the gene.

• Why is there a shift in the genes detected in your data? I assume this is before filtering, what is your QC metrics before and after filtering?

Response: Gene count displayed in Extended Data Figure 2b is after filtering. The shift is a direct consequence of lower coverage and artifacts removal in long reads (less UMI counts per cell).

• Aligned with the above two comments, why do you have so many genes detected and why do the numbers not reflect the greater number of genes detected in Extended data Figure 1b?

Response: We are not sure which figure or number the reviewer is referring to when asking “why do you have so many genes detected?”. If it refers to Extended Data Figure 2c, this is the total number

of genes detected overall (without counting novel genes) by both technologies, while **Extended data Figure 2b** indicates the distribution of the number of genes detected per-cell. As explained above, there was also a color inversion error in **Extended Data Figure 2**, which is now corrected. We hope this clarifies the question Reviewer 3 was raising.

Line 92-93:

- *This is not the case, there are references such as: PMID: 32887687, PMID: 36781734, PMID: 36289342*

Response: We appreciate the reviewer's comment and conducted a thorough investigation of these additional references in the context of those already present in our initial manuscript. In our manuscript, we stated that : “*So far, long-read RNA-seq has however only been applied on the bulk level in the field of oncology*”. The publications the reviewer is referring to are either a protocol for long-read scRNA-seq that does not involve the field of oncology⁵⁹ (PMID: 36781734), or the long-read sequencing is performed on bulk DNA-seq^{60,61} (PMID: 36289342, PMID: 32887687). To the best of our knowledge, there is currently no publication on patient-derived cancer samples using long-read scRNA-seq. Thus, we believe that the statement we have made in the original manuscript holds true. Yet, to address the reviewer’s request to tone down our statements, we have modified the sentence accordingly.

We have modified the sentence as follows:

“*Until now, long-read RNA-seq has primarily been utilized at the bulk level within the realm of oncology.*”

Line 102:

- *Please clarify this, as I believe there are other PacBio datasets that include more than 2,571 cells , e.g. this paper seems to have included 10,000 cells: <https://www.biorxiv.org/content/10.1101/2021.10.01.462818v>*

Response: Line 102-103: “*We applied both short-read and long-read scRNA-seq to five samples from three HGSOC patients, comprising 2,571 cells, and generated the largest PacBio scRNA-seq dataset to date*”. MAS-ISO-seq was applied to 6,260 CD8 T cells in a single run yielding 33 M reads corresponding to an overall lower sequencing depth - 1.5k UMIs/cell, compared to 2571 cells, total dataset size 212 M reads resulting in 30.7 M UMIs corresponding to 12k UMIs/cell. Our statement still holds true, this is the largest published PacBio scRNA-seq dataset to date in terms of number of reads and reads per cell. To avoid any confusion, we changed the line by:

“*We applied both short-read and long-read scRNA-seq to five samples from three HGSOC patients, comprising 2,571 cells, and generated the PacBio scRNA-seq dataset **with the deepest coverage** to date.*”

Figure 5:

- *It has been reported that template switching and PCR can generate significant false alternative transcripts and chimeric artefacts (<https://www.sciencedirect.com/science/article/pii/S0888754305003770>, <https://genomebiology.biomedcentral.com/articles/10.1186/gb-2011-12-2-r18>). You do not employ any strategy to determine that the IGF2BP2::TESPA1 is a real fusion event and not just a PCR artefact. I am not convinced that your novel fusion detection approach isn’t just detecting chimeric artefacts amplified early during the PCR cycles. No further validation is performed to confirm these fusion events. Your extended figure 5 states this this figure is intended as a validation of the IGF2BP2::TESPA1 breakpoint. However, this is just an IGV view and not an experimental validation.*
- *In order to be convinced, a strategy for removing chimeric artefacts is required or an experimental validation that does not include PCR.*

Response: We thank the reviewer for bringing up this point. We have performed several experiments in order to prove the presence of the fusion in the cancer patient. We are aware that PCR and RT-switching artefacts can lead to false alternative isoforms, which is why we used UMIs to remove PCR duplication effects. We used SQANTI classification to filter out RT-switching artifacts (see **Extended Data Figure 1c** in answer to Reviewer 3 above). In addition, all the reads are UMI-corrected.

We believe that the *IGF2BP2::TESPA1* fusion is present in the patient and not observed due to technical artifacts for the reasons presented below, where we first summarize the evidence presented in the original manuscript and then present additional evidence based on new experiments.

1. Evidence that *IGF2BP2::TESPA1* is highly unlikely to be a PCR / RT-switching artefact:

- The RNA fusion is patient 2-specific as it is not detected in the remaining patients (1 and 3) samples investigated. The fusion is covered by 2,174 UMIs in 178 cancer cells (**Figure 5c**, please refer to figures page 32-33 of this letter): a chimeric artefact amplified early during the PCR cycles would in principle only have one cell barcode and one UMI barcode.
- The fusion is observed in numerous RNA isoforms, featuring three distinct *IGF2BP2::TESPA1* exon-exon breakpoints. Importantly, all of these breakpoints adhere to canonical splice sites, displaying a seamless transition from an *IGF2BP2* exon to a *TESPA1* exon. As specified in the abstract of <https://www.sciencedirect.com/science/article/pii/S0888754305003770> cited above, RT template switching is found in non-canonical sites, and it less likely happens with canonical junctions: “However, most of the canonical examples (with ≥ 8 -nt direct repeats) that we found matched the majority (if not all) of the corresponding ESTs [expressed sequence tags, i.e. short cDNAs], indicating that they were genuine spliced isoforms. [...] These observations suggest that template-switching artifacts are rare among isoforms with canonical splice sites.”. Furthermore, in all 3 exon-exon breakpoints, there is no direct repeat of sequence that would suggest an RT-switching event.
- The detected fusion only uses a single *TESPA1* allele, while we identified two *TESPA1* alleles in non-cancer cells (using hSNPs, **Figure 5g**). Together with the patient and cancer cell specificity, a monoallelic fusion in a biallelic gene is supporting that *IGF2BP2::TESPA1* is indeed a biological alteration with genomic origin.
- Upon realignment, we observed that 93% of short-reads originally mapping to the *TESPA1* and *IGF2BP2* junction realigned on *IGF2BP2::TESPA1*, rather than the wild-type *IGF2BP2* or *TESPA1* (**Extended Data Figure 7a**). This means that, with prior knowledge of the existence of the fusion, it can also be detected in short reads, and is not a long-read artefact.
- We detected the fusion on the genomic level, in addition to the transcriptomic level, in an independent scDNA dataset of the same patient (**Figure 6, Extended Data Figure 7b** (previously **Extended Data Figure 5**)).

2. We aimed to further experimentally validate the fusion with additional experiments as follows:

- We have designed primer pairs in order to use genotyping PCR to detect the *IGF2BP2::TESPA1* fusion in genomic DNA isolated from multiple matched patient tumor samples. We have extracted genomic DNA from these patient samples, performed QC, and subjected 50 ng of genomic DNA to each PCR reaction. Our results clearly indicate that the fusion is present in multiple tumor samples derived from the same Patient 2 unlike in control samples derived from Patient 1 (**new Figure 6a**).
- In addition, we have purchased antibodies (antibodies information is below) in order to gain insights on the protein level of this particular gene fusion. We have also synthesized the expected fusion C-terminally tagged with a haemagglutinin (HA) tag (*IGF2BP2::TESPA1-HA*), cloned into pUltra plasmid (addgene #24129) for bicistronic expression of EGFP and the fusion HA, and lentivirally transduced cell lines HEK293 and ovarian cancer Kuramochi with the plasmid for verification of specificity of commercial antibodies (**Rebuttal Figure 2a,b**). Transfected cells were sorted based on the expressed GFP signal, indicating a >90% transfection rate for HEK293 and Kuramochi respectively (**Rebuttal Figure 2b**). Western blots revealed that HA antibodies detected a low expression at the expected fusion weight (~22kDa), mainly in HEK293 cells, confirming lentiviral transfection and plasmid expression (**Rebuttal Figure 2c**). Antibodies recognizing the C-terminal part of IGF2BP2 and TESPA1, used as negative controls (the TESPA1 part of the fusion is from the 3'UTR, and not part of the canonical TESPA1 protein), did not reveal a signal at the expected size overlapping with HA. Unfortunately, the AB recognizing the full-length IGF2BP2 protein did not show full or partial specificity towards the fusion, and only recognized wild-type IGF2BP2, thus not allowing any form of detection of the shortened fusion protein. In conclusion, we are confident based on the HA tag Western blot results that the fusion is expressed, however, the commercially available antibodies tested in this experimental setup did not recognize the fusion.
- Using the validated rabbit monoclonal antibody targeting C-terminal IGF2BP2, we detected a high wildtype IGF2BP2 protein level in Patient 2 compared to Patient 1 in matched tissue samples using immunofluorescence in combination with cancer cell detection and quantification (**Rebuttal Figure 2d,e**) supporting the differential *IGF2BP2* expression described in the results section. This suggests that the *IGF2BP2::TESPA1* fusion might have happened due to chromatin accessibility.
- Apart from confirming tumor-specific wildtype IGF2BP2 expression difference between patients on protein level that was observed on the gene expression level, our antibody setup does not allow us to draw conclusions on the protein expression of the patient-specific fusion. The full-length IGF2BP2-recognizing AB was not able to detect the shortened fusion protein and the lack of TESPA1 protein sequence in the fusion does not allow for detection of the fusion partner. Thus, the very nature of the shortened fusion product makes it difficult to detect on protein level with standard techniques.

Rebuttal Figure 2. Performance of commercially available antibodies to detect IGF2BP2 and TESPA1 in cell lines and patient samples. A) Principle of delivery of the fusion-encoding plasmid and EGFP⁺ enrichment after transduction in selected cell lines. **B)** Flow cytometry gating strategy with evaluation of EGFP⁺ cells after transduction and FACS-sorting for HEK293 and the ovarian cancer cell line Kuramochi. **C)** Immunodetection of overexpressed IGF2BP2::TESPA1-HA using various antibodies, red arrow indicates expected band for IGF2BP2 and TESPA1 antibodies. **D)** Representative immunofluorescence images (2 tissue regions per patient) for detection of tumor (EpCAM⁺) and C-terminal IGF2BP2 expressing cells in patient 1 (no fusion) and patient 2 harboring the IGF2BP2::TESPA1 fusion. **E)** Quantification of immunofluorescence images for determination of cancer cell-specific expression of IGF2BP2 in matched patient tissue samples,

Antibody details:

1. Mouse IGF2BP2 monoclonal antibody (MA5-25129) recognizing full length protein (aa 1-599)
2. Rabbit IGF2BP2 monoclonal antibody (MA5-42874) recognizing C-terminal peptide region (aa 500-595), which was truncated in the IGF2BP2::TESPA1 fusion
3. Rabbit TESPA1 polyclonal antibody (153352) recognizing 18 amino acid peptide near the amino terminus of human TESPA1
4. Rabbit HA monoclonal antibody (CST 3724) detects exogenously expressed proteins containing the HA epitope tag
5. Mouse monoclonal antibody GAPDH (sc-47724) used for reference protein expression
6. Mouse monoclonal antibody GFP (sc-9996) raised against aa 1-239 and used to show expression of EGFP in cell lines harbouring lentiviral plasmids

All antibodies have been tested on cell lines harbouring the overexpressed fusion together with EGFP as well as various patient samples depending on biobanked sample availability.

We have incorporated the obtained data on patient samples into the revised version of the manuscript. This encompasses the genotyping PCR results (**Figure 6a, see below**) as well as immunofluorescence (**Extended Data Figure 7**). Data obtained from the cell line experiments have only been included in the rebuttal letter, however, we are open to add them as supplementary data to the manuscript if the Reviewer feels that it would be appropriate. We also rewrote the 2nd paragraph of the “*Long-read sequencing captures gene fusions and identifies an IGF2BP2::TESPA1 fusion that was misidentified in short-read data*” section, as follows:

We next investigated the footprint of the gene fusion in the short-read data. The *TESPA1* gene was expressed in T cells, as well as in HGSOC cells, where its expression values were elevated. High expression was exclusive to patient 2 HGSOC cells and colocalized with *IGF2BP2* expression (**Fig. 5e,f**). *TESPA1* was the highest differentially expressed gene in cancer cells compared to non-cancer cells in patient 2 ($P_{\text{corr}} < 1.17 \times 10^{-14}$). Next, we re-aligned Patient 2's short-reads to a custom reference including the *IGF2BP2::TESPA1* transcriptomic breakpoint as well as wt *TESPA1* and wt *IGF2BP2* junctions (**Extended Data Fig. 7a, Methods**). Out of the 994 reads mapping to the custom reference, 93% preferentially aligned to *IGF2BP2::TESPA1* (99.8% of those were from HGSOC cells). **This means that, when given the option, reads previously aligning to *IGF2BP2* or *TESPA1* are preferentially mapping to the fusion reference, and the reported overexpression of *TESPA1* in short reads is likely an *IGF2BP2::TESPA1* expression.** Furthermore, reads covering the *TESPA1* 3' UTR region harbored three heterozygous single nucleotide polymorphisms (hSNPs): chr12:54.950.144 A>T (rs1047039), chr12:54.950.240 G>A (rs1801876), and chr12:54.950.349 C>G (rs2171497). In long-reads, wt *TESPA1* was either triple-mutated or not mutated at all, indicating two different alleles. All fusion long-reads, however, were triple-mutated, indicating a genomic origin and monoallelic expression of the fusion (**Fig. 5g**). In short reads, the three loci were mutated in nearly all reads, supporting the hypothesis that the observed *TESPA1* expression represents almost completely *IGF2BP2::TESPA1* expression and that it has a genomic origin.

Genomic breakpoint validation of the *IGF2BP2::TESPA1* fusion

To validate that the *IGF2BP2::TESPA1* gene fusion is the result of genomic rearrangements, both bulk and single-cell DNA sequencing (scDNA-seq) data from matched omental metastasis was used to query the genomic breakpoint. A putative genomic breakpoint was first found in the RNA data. Two long-read fusions were mapped to intronic regions of *IGF2BP2* and *TESPA1* (**Extended Data Fig. 7b**), pinpointing the location of the breakpoint at position chr3:185,694,020-chr12:54,960,603. **Subsequent genotyping PCR on genomic DNA extracted from patient-matched tissue samples using *IGF2BP2::TESPA1*, wt *IGF2BP2*, and wt *TESPA1* primer pairs flanking the genomic breakpoint confirmed the presence of *IGF2BP2::TESPA1* in Patient 2 in 3 out of 4 tested samples (**Fig. 6a, Methods**).** In contrast and as expected, the fusion was not found in Patient 1.

To assess whether the fusion was exclusive to cancer cells, we further investigated scDNA-seq data from Patient 2. For the identification of cancer cells, we inferred the scDNA-seq copy number profiles of all cells. We successfully identified two distinct clones within the pool of 162 cells. These clones encompassed a cancer clone designated as "Subclone 0" and a presumably non-cancer clone without copy number alterations labeled as "Subclone 1" (Fig. 6b**).** We next aligned the scDNA-seq data to a custom reference covering the breakpoint (**Methods**) and only found cancer reads mapping to the fusion breakpoint ($P=0.032$), while a mixture of reads from cancer and non-cancer cells mapped to wt

IGF2BP2 and wt *TESPA1* ($P=0.78$ and $P=1.00$, respectively) (**Fig. 6c**). Thus, genotyping PCR of bulk extracted DNA and scDNA-seq data confirmed the genomic fusion breakpoint in the intronic region detected in long-read scRNA-seq data. scDNA-seq also confirmed that the *IGF2BP2::TESPA1* fusion was cancer-cell specific, as suggested by long-read scRNA-seq data.

IGF2BP2 was also overexpressed in cancer cells from Patient 2 compared to other patients on both RNA and protein levels (**Extended Data Fig. 8a-c**). In Patient 2, there was an elevated copy number observed within the genomic region encompassing *IGF2BP2* (**Fig. 6b**). Therefore, the presence of a fusion allele on one allele does not seem to hinder the transcription of the wt *IGF2BP2* allele. Coherent with the high *IGF2BP2* protein levels, *IGF2* RNA, which is bound by the wt *IGF2BP2* protein, is also largely overexpressed in Patient 2 cancer cells compared to other patients (**Extended Data Fig. 8d**). This could indicate that the fusion happened partly due to accessible chromatin. In the ovarian cancer TCGA RNA dataset, the expression levels of exons surrounding breakpoints of *IGF2BP2* and *TESPA1* did not change (**Extended Data Fig. 9a,b**), and the overall expression of the genes did not correlate in any patient, suggesting that we detected an uncommon, patient-specific fusion (**Extended Data Fig. 9c**).

Figure 5: Tumor and patient-specific detection of novel *IGF2BP2::TESPA1* gene fusion.

(a) Overview of wt *IGF2BP2*, wt *TESPA1*, and *IGF2BP2::TESPA1* gene fusion with exon structure. **(b)** Overview of wt *IGF2BP2*, wt *TESPA1*, and fusion protein with protein domains. RRM: RNA-recognition motif, KH: heterogeneous nuclear ribonucleoprotein K-homology domain, KRAP_IP3R_bind: Ki-ras-induced actin-interacting protein-IP3R- interacting domain. **(c)** Violin plot showing patient- and tumor-specific *IGF2BP2::TESPA1* fusion transcript detection in patient 2. **(d)** UMI count in fusion-containing vs. -lacking patient 2 tumor cells. **(e)** UMAP embeddings of the cohorts' short-read data. Cells are colored if they express *IGF2BP2* (red), *TESPA1* (green), or both (yellow) in short- (left panel) or long-reads (right panel). **(f)** Raw expression of *TESPA1* (left) and *IGF2BP2* (right) in short- (top) or long-reads (bottom), by sample and cell type. **(g)** IGV view of short-reads (top), non-fusion long-reads (middle), and fusion long-reads (bottom) mapping to the 3'UTR of *TESPA1*. Non-fusion reads are either triple hSNP-mutated or non-mutated, while fusion and short-reads are only triple hSNP-mutated.

Figure 6: IGF2BP2::TESPA1 fusion breakpoint validation in bulk and scDNA.

(a) Genotyping PCR on genomic DNA isolated from matched patient samples using gene-specific primers for IGF2BP2::TESPA1 genomic breakpoint (top), wt TESPA1 (middle) and wt IGF2BP2 (bottom). **(b)** Copy number values per subclone in Patient 2 scDNA-seq data. Subclone 0 has multiple copy number alterations, indicative of cancer, while Subclone 1 is copy-number neutral, presumably non-cancer. **(c)** IGV view of scDNA reads aligning unambiguously to the IGF2BP2::TESPA1 genomic breakpoint (top), wt TESPA1 (middle), or wt IGF2BP2 (bottom). In red, reads from Subclone 0 cells (cancer); in blue, reads from Subclone 1 cells (non-cancer).

Extended Data Figure 8: Immunofluorescence validation of IGF2BP2 expression.

(a) Normalized expression of IGF2BP2 in cancer cells, per patient. **(b)** Representative immunofluorescence images (two tissue regions per patient) for detection of tumor (EpCAM⁺) and C-terminal IGF2BP2 expressing cells in patient 1 (no fusion) and patient 2 harboring the *IGF2BP2::TESPA1* fusion. **(c)** Quantification of immunofluorescence images for determination of cancer cell-specific expression of IGF2BP2 in matched patient tissue samples. **(d)** Normalized expression of IGF2 in cancer cells, per patient

In addition, we have added the following sections to the Material & Methods sections. Antibody details have been incorporated as well.

Genotyping PCR on genomic DNA

Genomic DNA was extracted from homogenized tumor tissue samples (n=8 samples matching sampling time, Basel Ovarian Biobank) derived from patients using QIAGEN DNeasy Blood & Tissue kit (#69504). Isolated DNA underwent QC using Nanodrop and Qubit measurements. Genotyping PCR on genomic DNA was performed using MyTaq DNA Polymerase system from Bioline. Briefly, MyTaq reaction buffer and MyTaq DNA polymerase were pipetted together with 200 nM primer pairs (gPCR_IGF2BP2-TESPA1_Bp_F 5'-CCT GCT TTG AGG AGG GGA GGG A-3' & gPCR_IGF2BP2-TESPA1_Bp_R 5'-ACT GAG GAC AAT GCT ACG CAA GA-3'; gPCR_TESPA1_F 5'-CCT GCT TTG AGG AGG GGA GGG A-3' & gPCR_TESPA1_R 5'-TGA GAA CTG CTG TTC CAG GAG ACA-3'; gPCR_IGF2BP2_F 5'-ACA CTG GAC CCA TGC TTG AGC T-3' & gPCR_IGF2BP2_R 5'-GCG TGC TAT GAA CAC TCC AGG CC-3'), and 50 ng genomic DNA (gDNA). PCR conditions were 1 cycle 94°C for 5 min followed by 35 cycles 95°C for 20 sec, 58°C for 15 sec, 72°C for 1 min and finished with 1 cycle at 72°C for 5 min. Amplicons were visualised on a 1.2 % agarose gel together with DNA ladder.

Immunofluorescence

Formalin-fixed and paraffin embedded tissue samples were obtained from the Basel Ovarian Biobank matching with patient 1 and 2 on sampling time and site. Briefly, samples were deparaffinized and immersed for 10 min in a 10 mM sodium citrate buffer at pH 6.0 (#C9999, Sigma Aldrich, Switzerland) at 95°C for antigen retrieval. Samples were permeabilized in 0.25% (v/v) Triton™ X-100 in PBS for 5 min and blocked in 5% FBS, 0.1% Triton™ X-100, 1% BSA in PBS for 1 hour. The following antibodies were used for this study: rabbit IGF2BP2 (#MA5-42874, ThermoFisher Scientific), EpCAM (#5488S, Cell Signaling Technologies) goat anti-rabbit Alexa Fluor® 647 (#4414, Cell Signaling Technology, Switzerland). Slides were mounted using ProLong® Gold Antifade Reagent with DAPI (#8961, Cell Signaling Technology, Switzerland). Images were acquired using a Nikon spinning-disk confocal microscope (Nikon CSU-W1 spinning-disk confocal microscope, Nikon Europe, Netherlands) and processed with Fiji. Cell quantification was performed using an in-house developed QuPath script for cell detection and annotations.

References

1. Mayr, C. & Bartel, D. P. Widespread shortening of 3'UTRs by alternative cleavage and polyadenylation activates oncogenes in cancer cells. *Cell* **138**, 673–684 (2009).
2. Williams, M., Cheng, Y. Y., Blenkiron, C. & Reid, G. Exploring mechanisms of microRNA downregulation in cancer. *Microna* **6**, 2–16 (2017).
3. Maurer, B. *et al.* MicroRNA-29, a key regulator of collagen expression in systemic sclerosis. *Arthritis Rheum.* **62**, 1733–1743 (2010).
4. Chen, Y. & Wang, X. miRDB: an online database for prediction of functional microRNA targets. *Nucleic Acids Res.* **48**, D127–D131 (2020).
5. Jin, Y., Song, X., Sun, X. & Ding, Y. Up-regulation of collagen type V alpha 2 (COL5A2) promotes malignant phenotypes in gastric cancer cell via inducing epithelial-mesenchymal transition (EMT). *Open Med (Wars)* **18**, 20220593 (2023).
6. Shintani, Y., Maeda, M., Chaika, N., Johnson, K. R. & Wheelock, M. J. Collagen I promotes epithelial-to-mesenchymal transition in lung cancer cells via transforming growth factor-beta signaling. *Am. J. Respir. Cell Mol. Biol.* **38**, 95–104 (2008).
7. Liberzon, A. *et al.* The Molecular Signatures Database (MSigDB) hallmark gene set collection. *Cell Syst.* **1**, 417–425 (2015).
8. Owusu-Ansah, K. G. *et al.* COL6A1 promotes metastasis and predicts poor prognosis in patients with pancreatic cancer. *Int. J. Oncol.* **55**, 391–404 (2019).
9. Ramadoss, S., Chen, X. & Wang, C.-Y. Histone demethylase KDM6B promotes epithelial-mesenchymal transition. *J. Biol. Chem.* **287**, 44508–44517 (2012).
10. Du, B. *et al.* The potential role of TNFAIP3 in malignant transformation of gastric carcinoma. *Pathol. Res. Pract.* **215**, 152471 (2019).
11. D'Arrigo, P. *et al.* The splicing FK506-binding protein-51 isoform plays a role in glioblastoma resistance through programmed cell death ligand-1 expression regulation. *Cell Death Discov.* **5**, 137 (2019).
12. Jie, W. *et al.* Pathophysiological functions of rnd3/rhoe. *Compr. Physiol.* **6**, 169–186 (2015).
13. Han, Q. *et al.* Omental cancer-associated fibroblast-derived exosomes with low microRNA-29c-3p promote ovarian cancer peritoneal metastasis. *Cancer Sci.* **114**, 1929–1942 (2023).
14. Chou, J. *et al.* GATA3 suppresses metastasis and modulates the tumour microenvironment by regulating microRNA-29b expression. *Nat. Cell Biol.* **15**, 201–213 (2013).
15. Smyth, A., Callaghan, B., Willoughby, C. E. & O'Brien, C. The Role of miR-29 Family in TGF-β Driven Fibrosis in Glaucomatous Optic Neuropathy. *Int. J. Mol. Sci.* **23**, (2022).

16. Yu, C.-C., Liao, Y.-W., Hsieh, P.-L. & Chang, Y.-C. Targeting lncRNA H19/miR-29b/COL1A1 Axis Impedes Myofibroblast Activities of Precancerous Oral Submucous Fibrosis. *Int. J. Mol. Sci.* **22**, (2021).
17. Tian, X., Zuo, X., Hou, M., Li, C. & Teng, Y. lncRNA-H19 regulates chemoresistance to carboplatin in epithelial ovarian cancer through microRNA-29b-3p and STAT3. *J. Cancer* **12**, 5712–5722 (2021).
18. Lv, M. *et al.* lncRNA H19 regulates epithelial-mesenchymal transition and metastasis of bladder cancer by miR-29b-3p as competing endogenous RNA. *Biochim. Biophys. Acta Mol. Cell Res.* **1864**, 1887–1899 (2017).
19. An, Q. *et al.* circKRT7-miR-29a-3p-COL1A1 Axis Promotes Ovarian Cancer Cell Progression. *Onco Targets Ther* **13**, 8963–8976 (2020).
20. Panda, A. C. Circular RNAs Act as miRNA Sponges. *Adv. Exp. Med. Biol.* **1087**, 67–79 (2018).
21. García-Bartolomé, A. *et al.* Altered Expression Ratio of Actin-Binding Gelsolin Isoforms Is a Novel Hallmark of Mitochondrial OXPHOS Dysfunction. *Cells* **9**, (2020).
22. Chen, Z.-Y., Wang, P.-W., Shieh, D.-B., Chiu, K.-Y. & Liou, Y.-M. Involvement of gelsolin in TGF-beta 1 induced epithelial to mesenchymal transition in breast cancer cells. *J. Biomed. Sci.* **22**, 90 (2015).
23. Liberzon, A. *et al.* Molecular signatures database (MSigDB) 3.0. *Bioinformatics* **27**, 1739–1740 (2011).
24. Cushing, L. *et al.* miR-29 is a major regulator of genes associated with pulmonary fibrosis. *Am. J. Respir. Cell Mol. Biol.* **45**, 287–294 (2011).
25. Feng, X. *et al.* TC3A: The Cancer 3' UTR Atlas. *Nucleic Acids Research.* **46**, 1027–1030 (2018).
26. Banday, A. R. *et al.* Genetic regulation of OAS1 nonsense-mediated decay underlies association with COVID-19 hospitalization in patients of European and African ancestries. *Nat. Genet.* **54**, 1103–1116 (2022).
27. Bertolini, A. *et al.* scAmpI-A versatile pipeline for single-cell RNA-seq analysis from basics to clinics. *PLoS Comput. Biol.* **18**, e1010097 (2022).
28. Germain, P.-L., Lun, A., Garcia Meixide, C., Macnair, W. & Robinson, M. D. Doublet identification in single-cell sequencing data using scDbIFinder. *F1000Res.* **10**, 979 (2021).
29. Hafemeister, C. & Satija, R. Normalization and variance stabilization of single-cell RNA-seq data using regularized negative binomial regression. *Genome Biol.* **20**, 296 (2019).
30. Levine, J. H. *et al.* Data-Driven Phenotypic Dissection of AML Reveals Progenitor-like Cells that Correlate with Prognosis. *Cell* **162**, 184–197 (2015).
31. Prummer, M. *et al.* scROSHI - robust supervised hierarchical identification of single cells. *BioRxiv* (2022) doi:10.1101/2022.04.05.487176.
32. Irmisch, A. *et al.* The Tumor Profiler Study: integrated, multi-omic, functional tumor profiling for clinical decision support. *Cancer Cell* **39**, 288–293 (2021).
33. Hao, Y. *et al.* Integrated analysis of multimodal single-cell data. *Cell* **184**, 3573–3587. (2021).
34. Newman, A. M. *et al.* Robust enumeration of cell subsets from tissue expression profiles. *Nat. Methods* **12**, 453–457 (2015).
35. Sade-Feldman, M. *et al.* Defining T Cell States Associated with Response to Checkpoint Immunotherapy in Melanoma. *Cell* **175**, 998–1013.e20 (2018).
36. Philippou, A., Maridakis, M., Pneumaticos, S. & Koutsilieris, M. The complexity of the IGF1 gene splicing, posttranslational modification and bioactivity. *Mol. Med.* **20**, 202–214 (2014).
37. Tagami, S., Eguchi, Y., Kinoshita, M., Takeda, M. & Tsujimoto, Y. A novel protein, RTN-XS, interacts with both Bcl-XL and Bcl-2 on endoplasmic reticulum and reduces their anti-apoptotic activity. *Oncogene* **19**, 5736–5746 (2000).
38. Gong, L. *et al.* RTN1-C mediates cerebral ischemia/reperfusion injury via ER stress and mitochondria-associated apoptosis pathways. *Cell Death Dis.* **8**, e3080 (2017).
39. Parreno, J., Amadeo, M. B., Kwon, E. H. & Fowler, V. M. Tropomyosin 3.1 association with actin stress fibers

is required for lens epithelial to mesenchymal transition. *Invest. Ophthalmol. Vis. Sci.* **61**, 2 (2020).

40. Nacu, S. *et al.* Deep RNA sequencing analysis of readthrough gene fusions in human prostate adenocarcinoma and reference samples. *BMC Med. Genomics* **4**, 11 (2011).
41. Rynne-Vidal, A. *et al.* Mesothelial-to-mesenchymal transition as a possible therapeutic target in peritoneal metastasis of ovarian cancer. *J. Pathol.* **242**, 140–151 (2017).
42. Cai, J. *et al.* Fibroblasts in omentum activated by tumor cells promote ovarian cancer growth, adhesion and invasiveness. *Carcinogenesis* **33**, 20–29 (2012).
43. Kenny, H. A. *et al.* Mesothelial cells promote early ovarian cancer metastasis through fibronectin secretion. *J. Clin. Invest.* **124**, 4614–4628 (2014).
44. Qin, W. *et al.* TGF- β /Smad3 signaling promotes renal fibrosis by inhibiting miR-29. *J. Am. Soc. Nephrol.* **22**, 1462–1474 (2011).
45. Yu, P.-N. *et al.* Downregulation of miR-29 contributes to cisplatin resistance of ovarian cancer cells. *Int. J. Cancer* **134**, 542–551 (2014).
46. Dasari, S., Fang, Y. & Mitra, A. K. Cancer associated fibroblasts: naughty neighbors that drive ovarian cancer progression. *Cancers (Basel)* **10**, (2018).
47. Li, H. *et al.* IGF-IR signaling in epithelial to mesenchymal transition and targeting IGF-IR therapy: overview and new insights. *Mol. Cancer* **16**, 6 (2017).
48. Liefers-Visser, J. A. L., Meijering, R. A. M., Reyners, A. K. L., van der Zee, A. G. J. & de Jong, S. IGF system targeted therapy: Therapeutic opportunities for ovarian cancer. *Cancer Treat. Rev.* **60**, 90–99 (2017).
49. Yang, Y. *et al.* Tumor Suppressor microRNA-138 Suppresses Low-Grade Glioma Development and Metastasis via Regulating IGF2BP2. *Onco Targets Ther* **13**, 2247–2260 (2020).
50. Wang, D. *et al.* Tspa1 is involved in late thymocyte development through the regulation of TCR-mediated signaling. *Nat. Immunol.* **13**, 560–568 (2012).
51. Dhamija, S. & Menon, M. B. Non-coding transcript variants of protein-coding genes - what are they good for? *RNA Biol.* **15**, 1025–1031 (2018).
52. Dhamija, S. & Diederichs, S. From junk to master regulators of invasion: lncRNA functions in migration, EMT and metastasis. *Int. J. Cancer* **139**, 269–280 (2016).
53. Joglekar, A., Foord, C., Jarroux, J., Pollard, S. & Tilgner, H. U. From words to complete phrases: insight into single-cell isoforms using short and long reads. *Transcription* 1–13 (2023) doi:10.1080/21541264.2023.2213514.
54. Lischetti, U. *et al.* Dynamic thresholding and tissue dissociation optimization for CITE-seq identifies differential surface protein abundance in metastatic melanoma. *Commun. Biol.* **6**, 830 (2023).
55. Fortelny, N., Overall, C. M., Pavlidis, P. & Freue, G. V. C. Can we predict protein from mRNA levels? *Nature* **547**, E19–E20 (2017).
56. Vogel, C. & Marcotte, E. M. Insights into the regulation of protein abundance from proteomic and transcriptomic analyses. *Nat. Rev. Genet.* **13**, 227–232 (2012).
57. Liu, Y., Beyer, A. & Aebersold, R. On the Dependency of Cellular Protein Levels on mRNA Abundance. *Cell* **165**, 535–550 (2016).
58. Tardaguila, M. *et al.* SQANTI: extensive characterization of long-read transcript sequences for quality control in full-length transcriptome identification and quantification. *Genome Res.* **28**, 396–411 (2018).
59. Philpott, M., Oppermann, U. & Cribbs, A. P. Long-Read Single-Cell Sequencing Using scCOLOR-seq. *Methods Mol. Biol.* **2632**, 259–267 (2023).
60. Funnell, T. *et al.* Single-cell genomic variation induced by mutational processes in cancer. *Nature* **612**, 106–115 (2022).
61. Sakamoto, Y. *et al.* Long-read sequencing for non-small-cell lung cancer genomes. *Genome Res.* **30**,

1243–1257 (2020).

REVIEWERS' COMMENTS

Reviewer #2 (Remarks to the Author):

The authors have addressed my concerns sufficiently in the revised manuscript.

Reviewer #3 (Remarks to the Author):

Thank you for the responses, I am now happy to recommend publication.

REVIEWERS' COMMENTS

Reviewer #2 (Remarks to the Author):

The authors have addressed my concerns sufficiently in the revised manuscript.

Reviewer #3 (Remarks to the Author):

Thank you for the responses, I am now happy to recommend publication.

Answer: We thank the reviewer for helping us ameliorate our manuscript throughout the revisions